

# Conformal partial waves in momentum space

**Marc Gillioz**

SISSA, via Bonomea 265, 34136 Trieste, Italy

## Abstract

The decomposition of 4-point correlation functions into conformal partial waves is a central tool in the study of conformal field theory. We compute these partial waves for scalar operators in Minkowski momentum space, and find a closed-form result valid in arbitrary space-time dimension $d \geq 3$ (including non-integer $d$). Each conformal partial wave is expressed as a sum over ordinary spin partial waves, and the coefficients of this sum factorize into a product of vertex functions that only depend on the conformal data of the incoming, respectively outgoing operators. As a simple example, we apply this conformal partial wave decomposition to the scalar box integral in $d = 4$ dimensions.

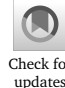

# 1 Introduction

Conformal field theory (CFT) can be defined non-perturbatively from its 2- and 3-point correlation functions, thanks to a convergent operator product expansion (OPE) that reduces the computation of higher-point functions to these elementary building blocks. In particular, the decomposition of 4-point functions into conformal partial waves is at the heart of the modern conformal bootstrap program. The bootstrap exploits the self-consistency of the OPE to put constraints on the *dynamics* of a theory encoded in the CFT data [1–4]. The problem of computing the conformal partial waves, i.e. determining the *kinematics* underlying 4-point correlators, has been solved long ago [5,6]. Closed-form results exist for scalar operators in dimensions $d = 2$ and 4, and various techniques for an efficient computation of the conformal partial waves in arbitrary dimensions have been developed, both for scalars [7–11] and for operators that carry spin [12–29].

In this work, we address a different but related problem: we compute the conformal partial waves in the momentum-space representation of correlation functions in Minkowski space, as opposed to the ordinary Euclidean position-space approach. We focus on correlation functions involving 4 scalar primary operators $\phi_i$, each carrying momentum $p_i$, in $d \geq 3$ dimensions, of the form

$$\langle 0|[\phi_1(p_1)\phi_2(p_2)][\phi_3(p_3)\phi_4(p_4)]|0\rangle \equiv (2\pi)^d \delta^d(p_1 + p_2 + p_3 + p_4) G(p_1, p_2, p_3), \quad (1.1)$$

where the square brackets around pairs of operators on the left-hand side can mean any of the following: no particular ordering (Wightman function), a commutator (possibly retarded or advanced), a time-ordered product or its hermitian conjugate. The conformal partial wave expansion for $G$ takes the form

$$G(p_1, p_2, p_3) = s^{(\Delta_1 + \Delta_2 + \Delta_3 + \Delta_4 - 3d)/2} \sum_{\mathcal{O}} \lambda_{12\mathcal{O}} \lambda_{\mathcal{O}34} G_{\Delta,\ell}(p_1, p_2, p_3), \quad (1.2)$$

where the sum is over primary operators $\mathcal{O}$ with scaling dimension $\Delta$ and spin $\ell$, $\lambda_{12\mathcal{O}}$ and $\lambda_{\mathcal{O}34}$ are OPE coefficients encoding the dynamical information about the theory, and $s$ is the center-of-mass energy

$$s = (p_1 + p_2)^2 = (p_3 + p_4)^2 > 0, \quad (1.3)$$

which is strictly positive by the spectral condition on the correlation function (1.1). The conformal partial waves $G_{\Delta,\ell}$ are completely fixed by symmetry in terms of $\Delta$ and $\ell$, of the scaling dimensions $\Delta_i$ of the external operators (which may be distinct or not) and of the space-time dimension $d$ (possibly including non-integer dimensions, as our results are analytic in $d \geq 3$). We obtain

$$G_{\Delta,\ell}(p_1, p_2, p_3) = \sum_{m=0}^{\ell} C_{\Delta,\ell,m} \mathcal{C}_m^{(d-3)/2}(\cos\theta) V_{\Delta,\ell,m}^{[12]}\left(\frac{p_1^2}{s}, \frac{p_2^2}{s}\right) V_{\Delta,\ell,m}^{[34]}\left(\frac{p_3^2}{s}, \frac{p_4^2}{s}\right), \quad (1.4)$$

where $\theta$ is the scattering angle, defined in terms of the momenta $p_i$ in eq. (2.10), and $\mathcal{C}_m^{(d-3)/2}(\cos\theta)$ a Gegenbauer polynomial of degree $m$ that appears in the ordinary spin partial wave expansion in $d$ dimensions. The numerical factor $C_{\Delta,\ell,m} \geq 0$ is given in eq. (2.40), and the "vertex functions" $V_{\Delta,\ell,m}^{[ab]}$ are defined in section 3, depending on which ordering is understood in the definition (1.1). This result admits a simple diagrammatic representation given in figure 1. One observes a factorization in the sense that the dependence on the external CFT data (the scaling dimensions $\Delta_i$) and on the "invariant masses" $p_i^2$ is entirely contained in the vertex functions $V_{\Delta,\ell,m}^{[ab]}$. Note that we have used the evident notation $p_4^2 = (p_1 + p_2 + p_3)^2$.

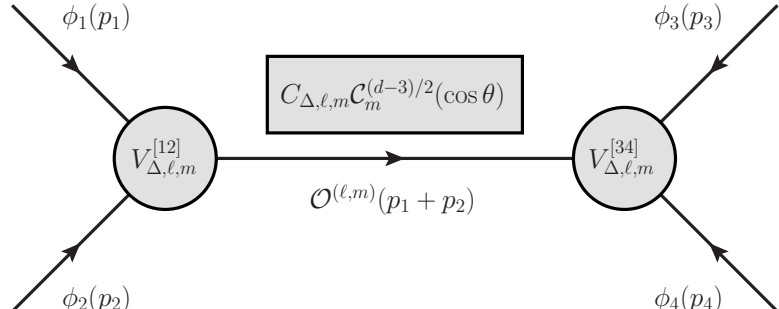

Figure 1: Diagrammatic representation of the conformal partial wave (1.4), in which each line corresponds to a local primary operator of the CFT carrying a certain momentum. Note that the vertex functions $V_{\Delta,\ell,m}^{[ab]}$ only depend on the scaling dimensions, on the spin (the total spin $\ell$ and its projection $m$ onto a reference direction), and on the momenta of the lines attached to it.

Sections 2 and 3 of this work are dedicated to the derivation of the momentum-space conformal partial waves (1.4). This derivation does not rely on solving a Casimir equation as in ordinary position space: in fact, since the conformal Ward identities are second-order differential equations with respect to the momenta $p_i$, one cannot make use of invariant cross-ratios in momentum space. Instead, the 4-point function depends on six Lorentz-invariant quantities (taken here to be all four $p_i^2$, $s$ and $\cos\theta$), and solving a differential equation in as many variables appears like a formidable task. This disadvantage is however counterbalanced by two elements that are unique to the Minkowski momentum-space representation of correlation functions:

- The orthogonality of momentum eigenstates, that allows to factorize the conformal partial wave into a product of 3-point functions, up to the contraction of Lorentz indices.

- The ability to use a reference frame, the center-of-mass frame, in which the 2-point function of the intermediate operator can be decomposed into polarizations that transform under irreducible representations of the rotation group $SO(d-1)$.

These points have been put forward in a recent derivation of position-space conformal blocks [24], as well as in Mack's classification of all the representations of the Lorentzian conformal group in $d = 4$ [30]. The operator product expansion in general and the conformal partial wave expansion (1.2) in particular are nothing but the use of a Hilbert space completeness relation, which in a CFT can be made explicit thanks to the two properties listed above. In fact, the correlation function (1.1) computes the overlap between two states of the theory: one state obtained by acting on the vacuum with the product of operators $[\phi_3\phi_4]$, and the other state obtained acting with $[\phi_1\phi_2]^\dagger$. The partial wave (1.4) corresponds to the projection of this overlap of states onto a conformal family consisting of a primary operator and of all of its descendants.

A corollary of this observation is that each conformal partial wave is positive in the forward scattering limit $p_1 \to -p_4$, $p_2 \to -p_3$ provided that $\phi_1 = \phi_4$ and $\phi_2 = \phi_3$. In this case the vertex functions are complex conjugate of each other, $V_{\Delta,\ell,m}^{[12]} = \left(V_{\Delta,\ell,m}^{[34]}\right)^*$, and the Gegenbauer polynomial evaluated at zero scattering angle ($\cos\theta = 1$) is positive, hence $G_{\Delta,\ell} \geq 0$. The positivity of this configuration and its decomposition into conformal partial waves have actually been used in the derivation of sum rules for anomaly coefficients [31,32].

It should be emphasized that the conformal partial wave expansion described in this work relies heavily on the Hamiltonian evolution being unitary, i.e. given by $\exp(iHt)$ where $t$ is the

Lorentzian time and $H$ a Hamiltonian operator bounded below. In a Euclidean theory, where this evolution is replaced by $\exp(-H\tau)$ with Euclidean "time" $\tau$, the OPE only makes sense for correlation functions that are time-ordered,[1] a property that is lost upon Fourier transforming. For this reason, the conformal partial wave expansion (1.2) differs at a fundamental level from other decompositions of the momentum-space 4-point function, such as the decomposition into Witten exchange diagrams in the case of a holographic theory (usually formulated in Mellin space [33–35]), or the decomposition into symmetric Polyakov blocks expressed directly in Euclidean momentum space [36, 37].[2] For the same reason, it is difficult to compare our conformal partial wave expansion with recent results on 4-point correlators in Euclidean momentum space [40–44].

On the other hand, eqs. (1.2) and (1.4) are immediately suitable for comparison with results obtained using ordinary Feynman diagram computations in theories that are both perturbative and conformal. While the goal of this work is not to study such situations systematically, we present in section 4 a simple example that involves the real part of the scalar box integral in four dimensions. This box integral appears in the 4-point function of the composite operator $\Phi^2$ in the theory of a free boson $\Phi$, or in that of the gauge-invariant operator $\mathrm{tr}(\Phi^i\Phi^i)$ in $\mathcal{N}=4$ supersymmetric Yang-Mills. The general form of this loop integral with off-shell momenta is known thanks to its dual conformal [45] or Yangian invariance [46].

Besides perturbative CFT, the conformal partial waves (1.4) will certainly find applications in other contexts. One of their salient features is the presence of double zeroes at scaling dimension of double-trace operators, a property reminiscent of the "double-twist" functional methods for the conformal bootstrap [47–52]. However, it should be emphasized that the correlator (1.1) is ill-suited for writing a bootstrap equation, as it does not have crossing symmetry built in. Microcausality in Minkowski momentum space implies certain analyticity properties of the correlation functions that might eventually be used to place bounds on the CFT data, but the relevant methods remain to be developed. Similarly, our conformal partial wave expansion could also find use in the cosmological bootstrap program [53–60], even though the connection between correlators in Minkowski momentum space and observables in de Sitter remains to be established (see ref. [61] for hints in this direction). For now, we provide the decomposition (1.2) as a tool and encourage the community to find its own applications.

## 2 Spin eigenstates and completeness relation

The core of the computation of the conformal partial waves is the formulation of a Hilbert space completeness relation. In this section, we assemble the different elements, construct a complete basis of states, and detail its dependence on the kinematic invariants of the 4-point function. The computation of the overlap of these states with the pairs of operators $[\phi_1\phi_2]$ and $[\phi_3\phi_4]$ is discussed instead in section 3.

### 2.1 Kinematics of the 4-point function

We begin with a detailed description of the kinematics of the 4-point function, as it will play a central role in what follows. As seen in eq. (1.1), momentum-space correlation functions

---

[1]If one uses radial quantization instead, then this applies to any configuration which is equivalent to a radially-ordered one under a global conformal symmetry transformation, which in practice extends to most of the space of configurations. The conformal bootstrap works precisely because it uses configurations that can be expanded into two or more distinct OPEs by different choices of quantization surfaces.

[2]Instead, there is an interesting connection between the conformal partial waves expansion described in this work and the Polyakov-Regge blocks recently introduced [38], through the generalization in momentum space of the concept of CFT dispersion relations [39]. We leave the exploration of this point for future work.

involving $n$ operators are always proportional to a delta function imposing momentum conservation, and hence depend non-trivially on $n-1$ momenta only. It is convenient to introduce the double-bracket notation

$$\langle 0 | \phi_1(p_1) \cdots \phi_n(p_n) | 0 \rangle \equiv (2\pi)^d \delta^d(p_1 + \ldots + p_n) \langle\!\langle \phi_1(p_1) \cdots \phi_n(p_n) \rangle\!\rangle . \tag{2.1}$$

Whenever this notation is being used, it is understood that one can trade one of the momenta for the sum of the others, say $p_n = -(p_1 + \ldots + p_{n-1})$.

For the 4-point function, this mean that there are 3 independent momenta, and hence 6 Lorentz invariants.[3] In analogy with scattering amplitudes, one can use the "invariant masses"

$$p_1^2, \quad p_2^2, \quad p_3^2, \quad p_4^2 = (p_1 + p_2 + p_3)^2 , \tag{2.2}$$

as well as the Mandelstam variables

$$s = (p_1 + p_2)^2, \qquad t = (p_1 + p_3)^2, \qquad u = (p_2 + p_3)^2 , \tag{2.3}$$

subject to the constraint $s + t + u = \sum p_i^2$. Unlike scattering amplitudes, however, we do not require the $p_i^2$ to take any special value. In fact, we even want to be agnostic about their sign: the momenta $p_i$ might be space-like, time-like, or even light-like. However, the specific ordering of the operators in eq. (1.1) does imply a constraint on the momentum flowing between the pairs $[\phi_1 \phi_2]$ and $[\phi_3 \phi_4]$: defining

$$p \equiv -p_1 - p_2 = p_3 + p_4 , \tag{2.4}$$

the correlation function (1.1) vanishes by the spectral condition whenever $p$ lies outside the forward light cone, i.e. unless

$$p^0 > 0, \qquad \text{and} \qquad s = p^2 > 0 . \tag{2.5}$$

We will assume henceforth that these two conditions are fulfilled.

With $s$ strictly positive, one can rescale all dimensionful quantities by $s$, including the 4-point function (1.1) whose overall scaling dimension is fixed by its operator content and by the space-time dimension. Since we chose in the definition (1.2) to factor out precisely the appropriate power of $s$, $G_{\Delta,\ell}$ is dimensionless. It is a function of the rescaled invariant masses

$$w_i \equiv \frac{p_i^2}{s} , \tag{2.6}$$

and of a fifth dimensionless variable that we define as follows: we introduce two new linear combinations of the momenta,

$$q_{12}^\mu \equiv \frac{(p \cdot p_2) p_1^\mu - (p \cdot p_1) p_2^\mu}{\sqrt{(p_1 \cdot p_2)^2 - p_1^2 p_2^2}}, \qquad\qquad q_{34}^\mu \equiv \frac{(p \cdot p_3) p_4^\mu - (p \cdot p_4) p_3^\mu}{\sqrt{(p_3 \cdot p_4)^2 - p_3^2 p_4^2}} , \tag{2.7}$$

which have the property of being both orthogonal to $p$, and normalized in units of $s$,

$$q_{12} \cdot p = q_{34} \cdot p = 0, \qquad\qquad q_{12}^2 = q_{34}^2 = -s . \tag{2.8}$$

The definitions (2.7) are non-singular as long as neither $p_1$ and $p_2$, nor $p_3$ and $p_4$ are colinear; the special configuration in which some of these momenta are colinear can be eventually

---

[3]We are always working in $d \geq 3$ dimensions. In $d = 2$, three vectors cannot be linearly independent and there is one less invariant

reached as a limit of the general case. We now define $\theta$ to be the angle between these two vectors,

$$\cos\theta \equiv -\frac{q_{12}\cdot q_{13}}{s}, \tag{2.9}$$

or in terms of the Mandelstam invariants

$$\cos\theta = \frac{s(u-t)-(p_1^2-p_2^2)(p_3^2-p_4^2)}{\sqrt{(s-p_1^2-p_2^2)^2-4p_1^2p_2^2}\sqrt{(s-p_3^2-p_4^2)^2-4p_3^2p_4^2}}. \tag{2.10}$$

$\theta$ corresponds in fact to the scattering angle in a $2 \to 2$ inelastic process in which the four "particles" have distinct masses (possibly including negative masses squared). In the center-of-mass frame, working for definiteness in $d = 3$, one can always choose

$$\begin{aligned}
p &= \sqrt{s}\,(1,0,0)\,,\\
q_{12} &= \sqrt{s}\,(0,1,0)\,,\\
q_{34} &= \sqrt{s}\,(0,\cos\theta,\sin\theta)\,,
\end{aligned} \tag{2.11}$$

which in terms of momenta $p_i$ corresponds to

$$\begin{aligned}
p_1 &= -\frac{\sqrt{s}}{2}\left(1+w_1-w_2,\Omega_{12},0\right)\,,\\
p_2 &= -\frac{\sqrt{s}}{2}\left(1-w_1+w_2,-\Omega_{12},0\right)\,,\\
p_3 &= \frac{\sqrt{s}}{2}\left(1+w_3-w_4,-\Omega_{34}\cos\theta,-\Omega_{34}\sin\theta\right)\,,
\end{aligned} \tag{2.12}$$

where we have defined

$$\Omega_{12} = \sqrt{(1-w_1-w_2)^2-4w_1w_2}, \qquad \Omega_{34} = \sqrt{(1-w_3-w_4)^2-4w_3w_4}. \tag{2.13}$$

The quantities under the square roots are non-negative for any configuration of momenta. Note that $\Omega_{12}$ and $\Omega_{34}$ will play an important role later. The center-of-mass configuration is illustrated in figure 2.

## 2.2 Polarization tensors

The Hilbert space completeness relation relies on a basis of momentum eigenstates that can be constructed out of a single primary operator insertion acting on the vacuum, for all primaries of the theory. In the case of scalar external operators, only scalar and traceless symmetric tensor operators enter the OPE. In order to avoid dealing with the contraction of tensor indices, and more importantly to obtain the factorized result (1.4), it is necessary to introduce a complete orthogonal basis of polarization tensors $\varepsilon_m^{\mu_1\dots\mu_\ell}$, after which one could write

$$\mathcal{O}^{\mu_1\dots\mu_\ell}(p)\,|0\rangle \equiv \sum_m \varepsilon_m^{\mu_1\dots\mu_\ell}\mathcal{O}^{(\ell,m)}(p)\,|0\rangle\,. \tag{2.14}$$

The goal of this section is to provide the explicit construction of such a basis, using only the momenta at hand, so that its transformation under special conformal transformations can later be determined explicitly. Such a construction is most easily done using spinor variables [62], but then it depends explicitly on the space-time dimension $d$. For scalar external operators, we will show that a $d$-independent construction can be performed.

The starting point is to realize that, thanks to the ability of going to the center-of-mass frame (2.11), the polarization tensors $\varepsilon_m^{\mu_1\dots\mu_\ell}$ can be decomposed into products of the vector

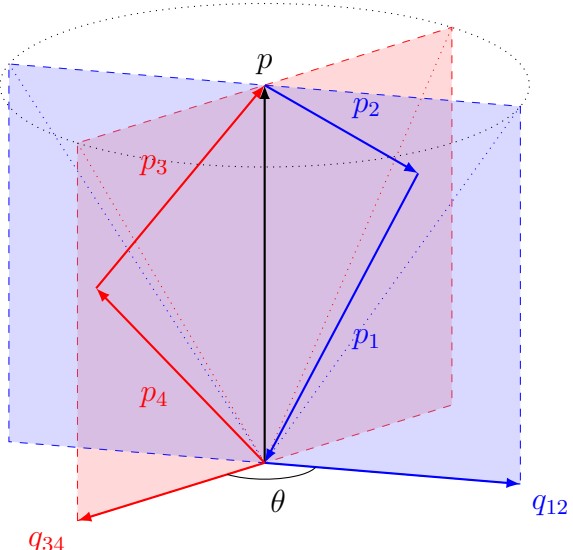

Figure 2: A possible configuration of momenta in the center-of-mass frame (energy along the vertical axis). The scattering angle $\theta$ corresponds to the angle between the planes spanned by $(p_1, p_2)$ and $(p_3, p_4)$, or equivalently between the vectors $q_{12}$ and $q_{34}$. In this particular example $p_1$ is time-like (backward directed), while $p_2$, $p_3$ and $p_4$ are all space-like, as seen by their position relative to the light cone (dotted lines); the covariant definition (2.10) is valid independently of the sign of the $p_i^2$.

$p$ and of invariant tensors of the group of spatial rotations $SO(d-1)$. An orthogonal basis of such polarization tensors is given by

$$\varepsilon_m^{\mu_1\cdots\mu_\ell} \equiv s^{-(\ell-m)/2}\left[\varepsilon_\perp^{(\mu_1\cdots\mu_m}p^{\mu_{m+1}}\cdots p^{\mu_\ell)} - \text{traces}\right], \tag{2.15}$$

where the indices on the right-hand side are symmetrized, and the tensors $\varepsilon_\perp^{\mu_1\cdots\mu_n}$ form themselves a basis of traceless, symmetric tensors orthogonal to $p$,

$$p_{\mu_1}\varepsilon_\perp^{\mu_1\cdots\mu_n} = 0. \tag{2.16}$$

This means that the tensors $\varepsilon_\perp^{\mu_1\cdots\mu_n}$ only have non-zero entries when all indices are spatial.

In our case, one can even go further and construct explicitly the tensors $\varepsilon_\perp^{\mu_1\cdots\mu_m}$ out of the momenta at hand. The reason is that we are only interested in polarization tensors that live in a plane spanned by a pair of momenta, say $p_1$ and $p_2$, or equivalently $p$ and $q_{12}$. Any polarization overlapping with a state created by two scalar operators acting on the vacuum is of this form. Using the projector onto the directions transverse to $p$,

$$\eta_\perp^{\mu\nu} \equiv \eta^{\mu\nu} - \frac{p^\mu p^\nu}{p^2}, \tag{2.17}$$

one has (see appendix A),

$$\varepsilon_\perp^{\mu_1\cdots\mu_m}(p,q) \equiv \sum_{n=0}^{\lfloor m/2 \rfloor} \frac{m!}{n!(m-2n)!} \frac{1}{2^{2n}\left(\frac{d-3}{2}+m-n\right)_n} \frac{\eta_\perp^{(\mu_1\mu_2}\cdots\eta_\perp^{\mu_{2n-1}\mu_{2n}}q^{\mu_{2n+1}}\cdots q^{\mu_m)}}{s^{m/2-n}}. \tag{2.18}$$

This is a covariant definition in terms of any pair of momenta $p$ and $q$ that satisfy $p \cdot q = 0$, $p^2 = -q^2 = s$. Similarly, a covariant definition of the more general tensors $\varepsilon_m^{\mu_1\cdots\mu_\ell}$ is directly

obtained plugging eq. (2.18) into (2.15). The final result is most easily expressed with the help of an auxiliary polarization vector $z^\mu$ satisfying $z^2 = 0$, in terms of which

$$z_{\mu_1} \cdots z_{\mu_\ell} \varepsilon_m^{\mu_1 \dots \mu_\ell}(p,q) = \frac{(z \cdot p)^{\ell - m}(z \cdot q)^m}{s^{\ell/2}} \, {}_2F_1\left(-\frac{m}{2}, -\frac{m-1}{2}; \frac{5-d}{2} - m; \frac{(z \cdot p)^2}{(z \cdot q)^2}\right). \quad (2.19)$$

Note that the hypergeometric series terminates both for even and odd $m$ (either one of the first two parameters is a negative integer), so that the right-hand side is in fact a homogeneous polynomial of degree $\ell$ in $z \cdot p$ and $z \cdot q$. The hypergeometric form is nevertheless convenient for its compactness, and it shows explicitly that the definition (2.19) is analytic in the space-time dimension $d$. It also gives a simple proof of the identities

$$\varepsilon_m^{\mu_1 \dots \mu_\ell}(p,q) = (-1)^m \varepsilon_m^{\mu_1 \dots \mu_\ell}(p,-q) = (-1)^{\ell - m} \varepsilon_m^{\mu_1 \dots \mu_\ell}(-p,q) = (-1)^\ell \varepsilon_m^{\mu_1 \dots \mu_\ell}(-p,-q). \quad (2.20)$$

The orthogonality of the basis follows by construction. Its normalization is given by

$$\varepsilon_{m' \mu_1 \dots \mu_\ell}(p,q) \varepsilon_m^{\mu_1 \dots \mu_\ell}(p,q) = \delta_{m'm}(-1)^m \mathcal{N}_{\ell,m}, \quad (2.21)$$

where

$$\mathcal{N}_{\ell,m} = \frac{m!(\ell - m)!}{\ell!} \frac{(d-2+2m)_{\ell-m}}{2^{\ell-m}\left(\frac{d-2}{2} + m\right)_{\ell-m}} \frac{(d-3)_m}{2^m \left(\frac{d-3}{2}\right)_m} \geq 0. \quad (2.22)$$

Moreover, if one considers two distinct polarization tensors defined with respect to two reference vectors $q$ and $q'$, both orthogonal to the same vector $p$, then one one obtains the identity (proof given in appendix A)

$$\varepsilon_{m' \mu_1 \dots \mu_\ell}(p,q) \varepsilon_m^{\mu_1 \dots \mu_\ell}(p,q') = \delta_{m'm}(-1)^m \mathcal{N}_{\ell,m} \frac{m!}{(d-3)_m} \mathcal{C}_m^{(d-3)/2}(\cos\theta), \quad (2.23)$$

where $\theta$ is the angle between $q$ and $q'$, and $\mathcal{C}_m^{(d-3)/2}$ is a Gegenbauer polynomial. This identity will play a central role in the computation of the conformal partial waves.

## 2.3 Spin eigenstates and normalization

We show next how these polarization tensors naturally appear in the 2-point function of primary operators, and how this leads to the construction of an orthogonal basis of spin eigenstates. The 2-point function of traceless, symmetric tensor operators in momentum space satisfies [63]

$$\langle\!\langle \mathcal{O}^{\mu_1 \dots \mu_\ell}(-p) \mathcal{O}^{\nu_1 \dots \nu_\ell}(p) \rangle\!\rangle = s^{\Delta - d/2} \Theta(p^0) \Theta(p^2) B_{\Delta,\ell} \sum_{n=0}^{\ell} \frac{(-1)^n 2^n \ell!}{n!(\ell-n)!} \frac{\left(\frac{d}{2} - \Delta\right)_n}{(2 - \Delta - \ell)_n}$$

$$\times \left[\frac{p^{(\mu_1} p^{\nu_1} \cdots p^{\mu_n} p^{\nu_n} \eta^{\mu_{n+1} \nu_{n+1}} \cdots \eta^{\mu_\ell \nu_\ell)}}{s^n} - \text{traces}\right], \quad (2.24)$$

where

$$B_{\Delta,\ell} = \frac{(4\pi)^{(d+2)/2} (\Delta - 1)_\ell}{2^{2\Delta+1} \Gamma\left(\Delta - \frac{d-2}{2}\right) \Gamma(\Delta + \ell)}. \quad (2.25)$$

This result is the unique solution (up to a normalization constant) to the conformal Ward identities for the 2-point function. Alternatively, it can be obtained from the direct Fourier transform of the position-space 2-point function in Lorentzian signature. The Heaviside $\Theta$-functions appearing in eq. (2.24) indicate that the 2-point function vanishes whenever $p$ lies outside the forward light cone.

To relate eq. (2.24) with the polarization tensors described above, we consider again its projection onto the plane spanned by $p$ and a vector $q$ orthogonal to it. This amounts to replacing

$$\eta^{\mu\nu} \to \frac{p^\mu p^\nu - q^\mu q^\nu}{s}, \tag{2.26}$$

in eq. (2.24). A bit of combinatorics (see appendix A) allows then to rewrite the 2-point function (2.24) as

$$\langle\!\langle \mathcal{O}^{\mu_1\cdots\mu_\ell}(-p)\mathcal{O}^{\nu_1\cdots\nu_\ell}(p) \rangle\!\rangle = s^{\Delta-d/2} \sum_{m=0}^{\ell} \frac{B_{\Delta,\ell,m}}{\mathcal{N}_{\ell,m}} \varepsilon_m^{\mu_1\cdots\mu_\ell}(-p,-q)\varepsilon_m^{\nu_1\cdots\nu_\ell}(p,q), \tag{2.27}$$

where

$$B_{\Delta,\ell,m} \equiv \frac{(\Delta-\ell-d+2)_{\ell-m}}{(\Delta+m-1)_{\ell-m}} B_{\Delta,\ell} \geq 0. \tag{2.28}$$

Requiring that the 2-point function defines a positive inner product, one actually recovers the well-known unitarity bounds $\Delta \geq (d-2)/2$ for scalars, and $\Delta - \ell \geq d-2$ for traceless, symmetric tensors of spin $\ell$. Note that in the generic case the 2-point function is a sum of $\ell + 1$ polarizations, the exception being spinning operators that saturate the unitarity bound: when $\Delta - \ell = d - 2$, all $B_{\Delta,\ell,m}$ with $m < \ell$ vanish: conserved currents only have transverse polarizations $m = \ell$.

This construction suggests to define the spin eigenstates

$$\big|\mathcal{O}^{(\ell,m)}(p\,|\,q)\big\rangle \equiv \mathcal{O}^{(\ell,m)}(p\,|\,q)\,|0\rangle \equiv \mathcal{N}_{\ell,m}^{-1}\varepsilon_m^{\mu_1\cdots\mu_\ell}(p,q)\mathcal{O}_{\mu_1\cdots\mu_\ell}(p)\,|0\rangle . \tag{2.29}$$

The notation indicates that this is a state carrying momentum $p$, with total spin $\ell$, and component $m$ along a direction that is defined by a reference vector $q$. These states form an orthogonal basis,

$$\big\langle \mathcal{O}^{(\ell',m')}(-p'\,|-q)\big|\mathcal{O}^{(\ell,m)}(p\,|\,q)\big\rangle = (2\pi)^d \delta^d(p'-p)\delta_{\ell'\ell}\delta_{m'm}\mathcal{N}_{\ell,m}^{-1}B_{\Delta,\ell,m}s^{\Delta-d/2}. \tag{2.30}$$

They realize the decomposition (2.14) envisioned at the beginning of this section. Note that an equivalent definition of the spin eigenstates can be obtained more generically by group-theoretical considerations, according to which $m$ labels the representation of the state under the little group that preserves the momentum $p$ [24, 30]. The advantage of our explicit construction in terms of the momentum $p$ and of the reference vector $q$ is that it is immediately suited to examine the transformation properties of the polarization tensors under special conformal transformations, as developed later in section 3.1.

## 2.4 Completeness relation

We are now equipped with a complete basis of states that can appear in the OPE of two scalar operators acting on the vacuum. This means that we can write

$$[\phi_3(p_3)\phi_4(p_4)]\,|0\rangle = \sum_{\mathcal{O}} \sum_{m=0}^{\ell} f_{\mathcal{O},m}(p_3,p_4)\big|\mathcal{O}^{(\ell,m)}(p_3+p_4\,|\,q_{34})\big\rangle, \tag{2.31}$$

where the sum is over all primary operators entering the OPE. To determine the coefficients $f_{\mathcal{O},m}(p_3,p_4)$, one computes the overlap of both sides of this equation with

$$\int_{\mathcal{V}_+} \frac{d^d k}{(2\pi)^d} \big\langle \mathcal{O}^{(\ell,m)}\big(-k\,|-q(k)\big)\big|, \tag{2.32}$$

where the integral is over the forward light cone ($k^2 > 0$ and $k^0 > 0$), and the reference vector is chosen as

$$q^\mu(k) \equiv \frac{k^2 q_{34}^\mu - (q_{34} \cdot k) k^\mu}{\sqrt{(q_{34} \cdot k)^2 - q_{34}^2 k^2}} . \tag{2.33}$$

This vector satisfies the two conditions $q(k) \cdot k = 0$ and $q(k)^2 = -k^2$ necessary to comply with our definition of the polarization tensors. Since (2.31) is a momentum eigenstate, the integral over $k$ collapses to a single value $k = p_3 + p_4 = p$, where $q(p) = q_{34}$, and one deduces that

$$f_{\mathcal{O},m}(p_3, p_4) = \frac{\mathcal{N}_{\ell,m}}{B_{\Delta,\ell,m}} s^{d/2-\Delta} \langle\!\langle \mathcal{O}^{(\ell,m)}(-p \,|\, -q_{34})[\phi_3(p_3)\phi_4(p_4)] \rangle\!\rangle . \tag{2.34}$$

The expression $\langle\!\langle \ldots \rangle\!\rangle$ is a function of $p_3$ and $p_4$ as it involves the 3-point function and a polarization tensor $\varepsilon_m^{\mu_1\ldots\mu_\ell}(-p, -q_{34})$ constructed out of the momenta $p_3$ and $p_4$ only.

The same logic applies to the product of scalar operators acting on the vacuum to the right, hence

$$\langle 0 | [\phi_1(p_1)\phi_2(p_2)] = \sum_{\mathcal{O}} \sum_{m=0}^{\ell} \frac{\mathcal{N}_{\ell,m}}{B_{\Delta,\ell,m}} s^{d/2-\Delta} \langle\!\langle [\phi_1(p_1)\phi_2(p_2)]\mathcal{O}^{(\ell,m)}(p \,|\, q_{12}) \rangle\!\rangle$$
$$\times \left\langle \mathcal{O}^{(\ell,m)}(-p \,|\, -q_{12}) \right| . \tag{2.35}$$

Using both eqs. (2.31) and (2.35) in the correlator (1.1) and resolving the trivial delta function, one gets

$$G(p_1, p_2, p_3) = \sum_{\mathcal{O}} \sum_{m=0}^{\ell} \left( \frac{\mathcal{N}_{\ell,m}}{B_{\Delta,\ell,m}} \right)^2 s^{d-2\Delta} \langle\!\langle \mathcal{O}^{(\ell,m)}(-p \,|\, -q_{12})\mathcal{O}^{(\ell,m)}(p \,|\, q_{34}) \rangle\!\rangle$$
$$\times \langle\!\langle [\phi_1(p_1)\phi_2(p_2)]\mathcal{O}^{(\ell,m)}(p \,|\, q_{12}) \rangle\!\rangle \langle\!\langle \mathcal{O}^{(\ell,m)}(-p \,|\, -q_{34})[\phi_3(p_3)\phi_4(p_4)] \rangle\!\rangle . \tag{2.36}$$

Note that we have also used the orthogonality of the 2-point function to write this expression as a single sum over primary operators $\mathcal{O}$ and as a sum over spin $m$. Using the representation (2.27) for the 2-point function and the property (2.23) of the polarization tensors, one can show that

$$\langle\!\langle \mathcal{O}^{(\ell,m)}(-p \,|\, -q_{12})\mathcal{O}^{(\ell,m)}(p \,|\, q_{34}) \rangle\!\rangle = s^{\Delta-d/2} \frac{B_{\Delta,\ell,m}}{\mathcal{N}_{\ell,m}} \frac{m!}{(d-3)_m} \mathcal{C}_m^{(d-3)/2}(\cos\theta) . \tag{2.37}$$

One arrives therefore at

$$G(p_1, p_2, p_3) = \sum_{\mathcal{O}} s^{d/2-\Delta} \sum_{m=0}^{\ell} C_{\Delta,\ell,m} \mathcal{C}_m^{(d-3)/2}(\cos\theta)$$
$$\times \langle\!\langle [\phi_1(p_1)\phi_2(p_2)]\mathcal{O}^{(\ell,m)}(p \,|\, q_{12}) \rangle\!\rangle \langle\!\langle \mathcal{O}^{(\ell,m)}(-p \,|\, -q_{34})[\phi_3(p_3)\phi_4(p_4)] \rangle\!\rangle . \tag{2.38}$$

This is now precisely in the form of eqs. (1.2) and (1.4), where the vertex functions are given by the 3-point correlators, and we have defined

$$C_{\Delta,\ell,m} = B_{\Delta,\ell,m}^{-1} \mathcal{N}_{\ell,m} \frac{m!}{(d-3)_m} , \tag{2.39}$$

or explicitly

$$C_{\Delta,\ell,m} = \frac{2^{2\Delta-\ell+1}(m!)^2(\ell-m)!(d-2+2m)_{\ell-m}\Gamma\left(\Delta - \frac{d-2}{2}\right)\Gamma(\Delta+\ell)}{(4\pi)^{(d+2)/2}\ell!\left(\frac{d-2}{2}+m\right)_{\ell-m}\left(\frac{d-3}{2}\right)_m(\Delta-1)_m(\Delta-\ell-d+2)_{\ell-m}} . \tag{2.40}$$

This coefficient diverges when $m < \ell$ and the unitarity bound $\Delta - \ell \geq d - 2$ is saturated. This is because we have included in the completeness relation (2.31) the null states corresponding to the longitudinal polarizations of conserved currents. We shall see however in section 3.3 that both of the 3-point functions entering eq. (2.38) vanish in this case. We can therefore safely exclude these null states from the sum.

Similarly, the coefficient $C_{\Delta,\ell,m}$ would be singular if one were to consider the identity operator ($\Delta = \ell = 0$) in the sum. Note however that the vacuum state does not contribute to the completeness relation (2.31) when $s > 0$, so it should be excluded as well.

Finally, the coefficient $C_{\Delta,\ell,m}$ is also singular in $d = 3$ for $m > 0$, but the Gegenbauer polynomial $C_m^{(d-3)/2}$ vanishes in the same case: the product of the two is in fact well-defined in the limit $d \to 3$, and it is proportional to the Chebyshev polynomial $T_m(\cos \theta)$.

In summary, using polarization tensors defined in terms of the kinematic variables, we have been able to write down a decomposition of the 4-point function into conformal partial waves, and to reduce the computation of these partial wave to a pair of 3-point functions, which we are going to discuss next.

# 3  Vertex functions

In this section we describe the computation of the 3-point functions appearing in eq. (2.38). This computations relies essentially on the results of ref. [64], but the projection onto spin eigenstates is new. At first this looks like a major complication, as the standard approach relying on solving conformal Ward identities requires to determine the transformation of the polarization tensors under special conformal symmetry. We shall see nonetheless that the results take a particularly simple form for the highest-spin function in each conformal family ($m = \ell$), and that the lower spin functions can then be determined recursively.

## 3.1  Conformal Ward identities

The starting point is to strip the 3-point correlation functions from its dynamical information, the OPE coefficient, and from an overall power of $s$ fixed by scaling symmetry. We write

$$\langle\!\langle [\phi_1(p_1)\phi_2(p_2)]\mathcal{O}^{(\ell,m)}(p \,|\, q_{12})\rangle\!\rangle \equiv \lambda_{12\mathcal{O}}\, s^{(\Delta_1+\Delta_2+\Delta-2d)/2} V_{\Delta,\ell,m}^{[12]}\left(\frac{p_1^2}{s}, \frac{p_2^2}{s}\right). \qquad (3.1)$$

$V_{\Delta,\ell,m}^{[12]}$ on the right-hand side is a function of two dimensionless variables only. It is precisely the vertex function that enters in the conformal partial wave (1.4). All information about scale and Poincaré symmetry is already included in this equation and in the definition (2.19) of the polarization tensors. What remains to be determined is actually contained in the action of special conformal transformations, together with a boundary condition that can be traced back to Euclidean position space. Note that the vertex function $V_{\Delta,\ell,m}^{[34]}$ in eq. (1.4) is simply related to $V_{\Delta,\ell,m}^{[12]}$ by hermitian conjugation and a change in the labels $1 \to 4$ and $2 \to 3$. We shall therefore focus exclusively on $V_{\Delta,\ell,m}^{[12]}$ in this whole section.

Working in Minkowski space, there is no choice but to use the infinitesimal form of special conformal transformation, as their exponentiated form does not preserve the causal structure of space-time. At the level of correlation functions, this means solving a partial differential equation: the Ward identity for special conformal transformations in momentum space is a second-order differential equation that can be written as [65, 66]

$$\widehat{K}_{12}^\alpha \langle\!\langle [\phi_1(p_1)\phi_2(p_2)]\mathcal{O}^{\mu_1\cdots\mu_\ell}(-p_1-p_2)\rangle\!\rangle = 0, \qquad (3.2)$$

where $\widehat{K}_{12}^\alpha$ is the differential operator[4]

$$\widehat{K}_{12}^\alpha = \sum_{i=1}^{2} \left[ p_i^\beta \frac{\partial^2}{\partial p_{i\alpha} \partial p_i^\beta} - \frac{1}{2} p_i^\alpha \frac{\partial^2}{\partial p_{i\beta} \partial p_i^\beta} - (\Delta_i - d) \frac{\partial}{\partial p_{i\alpha}} \right].$$  (3.3)

Note that this differential equation is valid no matter what the ordering between the operators $\phi_1$ and $\phi_2$ is: the difference in orderings only means different boundary conditions, which will be discussed later in section 3.2.

The Ward identity (3.2) does not apply directly to the vertex function $V_{\Delta,\ell,m}^{[12]}$, but rather to the correlator with the primary operator $\mathcal{O}^{\mu_1\cdots\mu_\ell}$ acting on the vacuum. In order to include the polarization tensors and the power of $s$ in this equation, we write, in accordance with eq. (2.14),

$$\langle\!\langle [\phi_1(p_1)\phi_2(p_2)] \mathcal{O}^{\mu_1\cdots\mu_\ell}(-p_1-p_2) \rangle\!\rangle = \lambda_{12\mathcal{O}} \, s^{(\Delta_1+\Delta_2+\Delta-2d)/2} \sum_{m=0}^{\ell} \varepsilon_m^{\mu_1\cdots\mu_\ell}(p,q_{12}) V_{\Delta,\ell,m}^{[12]}(w_1,w_2).$$  (3.4)

We also introduce an external polarization vector $z^\mu$ as in eq. (2.19), and use the shorthand notation

$$\varepsilon_m \equiv z_{\mu_1} \cdots z_{\mu_\ell} \varepsilon_m^{\mu_1\cdots\mu_\ell}(p,q_{12}).$$  (3.5)

It is straightforward (if tedious) to compute the action of $\widehat{K}_{12}^\alpha$ on the polarization tensors, as they are explicitly defined in terms of $p_1$ and $p_2$. In doing so, we encounter derivatives of the hypergeometric function (2.19), and we find it convenient to introduce

$$\varepsilon_m' \equiv \frac{\partial \varepsilon_m}{\partial(z \cdot q_{12})},$$  (3.6)

which describes a traceless symmetric tensor with $\ell - 1$ indices. The Ward identity (3.2) is now a simple vector equation, whose components can be resolved taking its scalar product with various reference vectors. For instance, the Ward identity given by $p \cdot \widehat{K}_{12}$ becomes

$$\sum_{m=0}^{\ell} \varepsilon_m \left\{ \Omega_{12}^m \left[ (1+w_1-w_2)\widehat{D}_1 + (1-w_1+w_2)\widehat{D}_2 \right] \Omega_{12}^{-m} \right. \\ \left. + \frac{(\ell-m)(d-2+\ell+m)}{2} \right\} V_{\Delta,\ell,m}^{[12]}(w_1,w_2) = 0.$$  (3.7)

If one introduce an auxiliary vector $r$ orthogonal to both $p$ and $q_{12}$, the Ward identity $r \cdot \widehat{K}_{12}$ gives

$$\sum_{m=0}^{\ell} \left\{ \varepsilon_m' \Omega_{12}^{-1} \widehat{D}_0 + \varepsilon_{m-1}' \frac{m(\Delta-m-d+2)(d-3+\ell+m)}{(d-3+2m)(d-5+2m)} \right. \\ \left. - \varepsilon_{m+1}' \frac{(\ell-m)(\Delta+m-1)}{m+1} \right\} V_{\Delta,\ell,m}^{[12]}(w_1,w_2) = 0.$$  (3.8)

---

[4]This operator only involves the scaling dimensions $\Delta_1$ and $\Delta_2$ of the scalar operators $\phi_1$ and $\phi_2$, and not that of $\mathcal{O}$; coincidentally, it does not act on the spin indices of the operator $\mathcal{O}$. This is a consequence of translation symmetry: in position space, one can use this symmetry to place $\mathcal{O}$ at the origin of the coordinate system, where the action of the special conformal transformations is trivial; in momentum space, once the delta function imposing momentum conservation is factored out of the 3-point function, $\widehat{K}_{12}^\alpha$ only involves the conformal data of two out of three operators.

Finally, the remaining content of the Ward identity can be obtained from the scalar product $q_{12} \cdot \widehat{K}_{12}$, which after subtraction of eq. (3.8) leads to

$$\sum_{m=0}^{\ell} \left\{ \varepsilon_m \Omega_{12}^m \left( \widehat{D}_1 - \widehat{D}_2 \right) \Omega_{12}^{-m} \right.$$
$$\left. - \varepsilon'_m \frac{z \cdot p}{\Omega_{12}} (\Delta - \ell - d + 2) - \varepsilon'_{m-1} \frac{z \cdot p}{\Omega_{12}^2} \frac{m}{d - 5 + 2m} \widehat{D}_0 \right\} V_{\Delta,\ell,m}^{[12]}(w_1, w_2) = 0 . \qquad (3.9)$$

Several elements of eqs. (3.7), (3.8) and (3.9) must still be explained: the quantity $\Omega_{12} = \sqrt{(1 - w_1 - w_2)^2 - 4 w_1 w_2}$ was originally defined in eq. (2.13); $\widehat{D}_1$ and $\widehat{D}_2$ are elliptic differential operators of the Appell $F_4$ type,

$$\widehat{D}_1 = w_1(1 - w_1)\frac{\partial^2}{\partial w_1^2} - 2 w_1 w_2 \frac{\partial^2}{\partial w_1 \partial w_2} - w_2^2 \frac{\partial^2}{\partial w_2^2}$$
$$+ \left[ c_1 - (a + b + 1) w_1 \right] \frac{\partial}{\partial w_1} - (a + b + 1) w_2 \frac{\partial}{\partial w_2} - a b , \qquad (3.10)$$

$$\widehat{D}_2 = w_2(1 - w_2)\frac{\partial^2}{\partial w_2^2} - 2 w_1 w_2 \frac{\partial^2}{\partial w_1 \partial w_2} - w_1^2 \frac{\partial^2}{\partial w_1^2}$$
$$+ \left[ c_2 - (a + b + 1) w_2 \right] \frac{\partial}{\partial w_2} - (a + b + 1) w_1 \frac{\partial}{\partial w_1} - a b , \qquad (3.11)$$

with

$$a = \frac{d + \Delta - \Delta_1 - \Delta_2 + m}{2} , \qquad\qquad c_1 = 1 + \frac{d}{2} - \Delta_1 ,$$
$$b = \frac{2d - \Delta - \Delta_1 - \Delta_2 + m}{2} , \qquad\qquad c_2 = 1 + \frac{d}{2} - \Delta_2 ; \qquad (3.12)$$

the remaining differential operator $\widehat{D}_0$ is given by

$$\widehat{D}_0 = (1 - w_1 + w_2)\left[ -2 w_1 \frac{\partial}{\partial w_1} + (\Delta_1 - d + 1) \right]$$
$$- (1 + w_1 - w_2)\left[ -2 w_2 \frac{\partial}{\partial w_2} + (\Delta_2 - d + 1) \right] . \qquad (3.13)$$

The interpretation of this system of partial differential equations for $\ell + 1$ functions of 2 variables $w_1$ and $w_2$ appears difficult at first. This is worsened by the fact that we have not expressed the conformal Ward identities in the complete basis $\{\varepsilon_m\}$ of traceless, symmetric tensors, but have instead appealed to the additional tensors $\varepsilon'_m$. Nevertheless, the system contains a lot of redundancies, due in particular to the closure of the conformal algebra in momentum space, and we will see in the next sections how a simple solution emerges.

## 3.2 Highest-spin functions in various orderings

The first key observation to be made is that the Ward identities (3.7) and (3.9) contain a closed system of equations for the function $V_{\Delta,\ell,\ell}^{[12]}(w_1, w_2)$, i.e. the highest-spin component $m = \ell$. The tensors $(z \cdot p)\varepsilon'_i$ appearing in eq. (3.9) are orthogonal to $\varepsilon_\ell$, the latter being totally transverse while the former not. When projecting against $\varepsilon_\ell$, these two equations give therefore the system

$$\left[ (1 + w_1 - w_2)\widehat{D}_1 + (1 - w_1 + w_2)\widehat{D}_2 \right] \Omega_{12}^{-\ell} V_{\Delta,\ell,\ell}^{[12]}(w_1, w_2) = 0 , \qquad (3.14)$$

$$\left( \widehat{D}_1 - \widehat{D}_2 \right) \Omega_{12}^{-\ell} V_{\Delta,\ell,\ell}^{[12]}(w_1, w_2) = 0 , \qquad (3.15)$$

which is equivalent to

$$\widehat{D}_1\left[\Omega_{12}^{-\ell}V^{[12]}_{\Delta,\ell,\ell}(w_1,w_2)\right]=\widehat{D}_2\left[\Omega_{12}^{-\ell}V^{[12]}_{\Delta,\ell,\ell}(w_1,w_2)\right]=0\,. \tag{3.16}$$

As already stated, $\widehat{D}_1$ and $\widehat{D}_2$ form a system of generalized hypergeometric differential equations of the Appell $F_4$ type, with a particular solution being $F_4(a,b;c_1,c_2;w_1,w_2)$. General solutions to such systems can be constructed from linear combinations of the four functions [67]

$$w_1^{\Delta_1-d/2}w_2^{\Delta_2-d/2}F_{\Delta_1\Delta_2;\Delta,\ell}(w_1,w_2), \qquad\qquad w_1^{\Delta_1-d/2}F_{\widetilde{\Delta}_1\Delta_2;\Delta,\ell}(w_1,w_2),$$
$$w_2^{\Delta_2-d/2}F_{\Delta_1\widetilde{\Delta}_2;\Delta,\ell}(w_1,w_2), \qquad\qquad F_{\widetilde{\Delta}_1\widetilde{\Delta}_2;\Delta,\ell}(w_1,w_2), \tag{3.17}$$

where we have used $\widetilde{\Delta}_i=d-\Delta_i$ and introduced the following notation for the Appell $F_4$ function:

$$F_{\Delta_a\Delta_b;\Delta,\ell}(w_1,w_2)\equiv F_4\left(\begin{array}{c}\frac{\Delta_a+\Delta_b-\Delta+\ell}{2},\ \frac{\Delta_a+\Delta_b+\Delta+\ell-d}{2}\\[4pt]\Delta_a-\frac{d}{2}+1,\ \Delta_b-\frac{d}{2}+1\end{array};w_1,w_2\right). \tag{3.18}$$

Note that the function $F$ satisfies the identities

$$F_{\Delta_1\Delta_2;\Delta,\ell}(w_1,w_2)=F_{\Delta_1\Delta_2;\widetilde{\Delta},\ell}(w_1,w_2)=F_{\Delta_2\Delta_1;\Delta,\ell}(w_2,w_1)\,. \tag{3.19}$$

An equivalent basis of solutions is obtained after the change of variables $(w_1,w_2)\to(w_1/w_2,1/w_2)$, in terms of the four functions

$$w_1^{\Delta_1-d/2}w_2^{(\Delta_2-\Delta_1-\Delta-\ell)/2}F_{\Delta_1\Delta;\Delta_2,\ell}\left(\frac{w_1}{w_2},\frac{1}{w_2}\right), \qquad w_2^{(\Delta_1+\Delta_2-\Delta-\ell-d)/2}F_{\widetilde{\Delta}_1\Delta;\Delta_2,\ell}\left(\frac{w_1}{w_2},\frac{1}{w_2}\right),$$
$$w_1^{\Delta_1-d/2}w_2^{(\Delta_2-\Delta_1+\Delta-\ell-d)/2}F_{\Delta_1\widetilde{\Delta};\Delta_2,\ell}\left(\frac{w_1}{w_2},\frac{1}{w_2}\right), \qquad w_2^{(\Delta_1+\Delta_2+\Delta-\ell-2d)/2}F_{\widetilde{\Delta}_1\widetilde{\Delta};\Delta_2,\ell}\left(\frac{w_1}{w_2},\frac{1}{w_2}\right). \tag{3.20}$$

The highest-spin vertex function $V^{[12]}_{\Delta,\ell,\ell}(w_1,w_2)$ is therefore a linear combination of the functions (3.17), or of the functions (3.20), multiplied by $\Omega_{12}^\ell$. What linear combination depends on the ordering of the operators in the 3-point function (3.1). We examine below two distinct cases.

**Wightman function**   When $\phi_1$ and $\phi_2$ are out-of-time-order, the Wightman function (3.1) has been computed in ref. [64] (see also ref. [68]). The starting point is the Euclidean position-space 3-point function

$$z_{\mu_1}\cdots z_{\mu_\ell}\,\langle0|\,\phi_1(x_1)\phi_2(x_2)\mathcal{O}^{\mu_1\cdots\mu_\ell}(0)\,|0\rangle$$
$$=\frac{\lambda_{12\mathcal{O}}\left[x_2^2(z\cdot x_1)-x_1^2(z\cdot x_2)\right]^\ell}{|x_1|^{\Delta_1-\Delta_2+\Delta+\ell}|x_2|^{\Delta_2-\Delta_1+\Delta+\ell}|x_1-x_2|^{\Delta_1+\Delta_2-\Delta+\ell}}\,. \tag{3.21}$$

This equation specifies our normalization of the OPE coefficient $\lambda_{12\mathcal{O}}$. If we define the Lorentzian 3-point function by analytic continuation and take its Fourier transform, we arrive at [64]

$$z_{\mu_1}\cdots z_{\mu_\ell}\langle\!\langle\phi_1(p_1)\phi_2(p_2)\mathcal{O}^{\mu_1\cdots\mu_\ell}(p)\rangle\!\rangle$$
$$=\lambda_{12\mathcal{O}}\frac{(-i)^\ell(4\pi)^{d+2}2^{-(\Delta_1+\Delta_2+\Delta+\ell+2)}(\Delta-1)_\ell}{\Gamma\left(\Delta_1-\frac{d}{2}+1\right)\Gamma\left(\Delta-\frac{d}{2}+1\right)\Gamma\left(\frac{\Delta_2-\Delta_1+\Delta+\ell}{2}\right)\Gamma\left(\frac{\Delta_1+\Delta_2-\Delta+\ell}{2}\right)}$$
$$\times\frac{(p^2)^{\Delta-d/2}(p_1^2)^{\Delta_1-d/2}}{(-p_2^2)^{(\Delta_1-\Delta_2+\Delta+\ell)/2}}\sum_{n=0}^{\ell}(z\cdot p_1)^{\ell-n}(z\cdot p)^n\mathcal{F}_n\left(\frac{p_1^2}{p_2^2},\frac{p^2}{p_2^2}\right), \tag{3.22}$$

where

$$\mathcal{F}_0\left(\frac{w_1}{w_2}, \frac{1}{w_2}\right) = F_{\Delta_1\Delta;\Delta_2,\ell}\left(\frac{w_1}{w_2}, \frac{1}{w_2}\right), \tag{3.23}$$

and the $\mathcal{F}_n$ with $n > 0$ are obtained acting recursively on $\mathcal{F}_0$ with a certain differential operator whose form is not important in our discussion: as we shall see, $\mathcal{F}_0$ is all we need to determine the highest-spin vertex function. Eq. (3.22) should be matched with our own formalism, in which[5]

$$z_{\mu_1}\cdots z_{\mu_\ell}\langle\!\langle\phi_1(p_1)\phi_2(p_2)\mathcal{O}^{\mu_1\cdots\mu_\ell}(p)\rangle\!\rangle$$
$$= \lambda_{12\mathcal{O}}\,(p^2)^{(\Delta_1+\Delta_2+\Delta-2d)/2}\sum_{m=0}^{\ell}z_{\mu_1}\cdots z_{\mu_\ell}\varepsilon_m^{\mu_1\cdots\mu_\ell}(p,q_{12})V_{\Delta,\ell,m}^{12}\left(\frac{p_1^2}{p^2},\frac{p_2^2}{p^2}\right). \tag{3.24}$$

If we consider this to be a function of $p$ and $p_1$, taking

$$p_2 = -p - p_1, \qquad q_{12} = \frac{(p\cdot p_1)p^\mu - (p^2)p_1^\mu}{\sqrt{(p\cdot p_1)^2 - p^2 p_1^2}}, \tag{3.25}$$

and if we expand the polarization tensors into powers of $z\cdot p_1$ using eq. (2.19), we obtain

$$z_{\mu_1}\cdots z_{\mu_\ell}\langle\!\langle\phi_1(p_1)\phi_2(p_2)\mathcal{O}^{\mu_1\cdots\mu_\ell}(p)\rangle\!\rangle$$
$$= \lambda_{12\mathcal{O}}\,(p^2)^{(\Delta_1+\Delta_2+\Delta+\ell-2d)/2}\frac{(-1)^\ell(z\cdot p_1)^\ell}{\left[(p\cdot p_1)-p^2 p_1^2\right]^{\ell/2}}V_{\Delta,\ell,\ell}^{12}\left(\frac{p_1^2}{p^2},\frac{(p+p_1)^2}{p^2}\right) + \dots, \tag{3.26}$$

where the ellipsis indicate lower powers of $z\cdot p_1$ (hence higher powers of $z\cdot p$), which are more complicated as they involve linear combinations of $V_{\Delta,\ell,m}^{12}$ with distinct $m$. The leading power in $z\cdot p_1$ must match with the term $n = 0$ in eq. (3.22), implying

$$\boxed{\begin{aligned}V_{\Delta,\ell,\ell}^{12}(w_1, w_2) &= \frac{i^\ell(4\pi)^{d+2}2^{-(\Delta_1+\Delta_2+\Delta+\ell+2)}(\Delta-1)_\ell\left[(1-w_1-w_2)^2-4w_1 w_2\right]^{\ell/2}}{\Gamma\left(\Delta_1-\frac{d}{2}+1\right)\Gamma\left(\Delta-\frac{d}{2}+1\right)\Gamma\left(\frac{\Delta_2-\Delta_1+\Delta+\ell}{2}\right)\Gamma\left(\frac{\Delta_1+\Delta_2-\Delta+\ell}{2}\right)}\\ &\times w_1^{\Delta_1-d/2}(-w_2)^{(\Delta_2-\Delta_1-\Delta-\ell)/2}F_{\Delta_1\Delta;\Delta_2,\ell}\left(\frac{w_1}{w_2},\frac{1}{w_2}\right).\end{aligned}} \tag{3.27}$$

It is remarkable that this vertex function is given by a single Appell $F_4$ function out of the four solutions (3.20). This has to do with the observation made in ref. [64] that the Euclidean OPE fixes the behavior of the Wightman 3-point function in a neighborhood of $p_1^2 = p^2 = 0$, i.e. around $w_1 = 0$ and $w_2 \to -\infty$.

The result (3.27) is valid when $p_1$ is time-like and backward-directed (the Wightman function vanishes otherwise by the spectral condition), and when $p_2$ is space-like. The case of time-like $p_2$ can be obtained in terms of the functions in the list (3.17) following the analysis of ref. [64].

**Time-ordered product** In the case where the 3-point function (3.1) contains a time-ordered product of the operators $\phi_1$ and $\phi_2$, the general structure is more complicated, as both $p_1$ and $p_2$ can be either space-like or time-like. We focus here on the case where they are both space-like, i.e. $w_1, w_2 < 0$. In this case, the Euclidean OPE condition only gives information

---

[5]Note that we replaced the generic label [12] with 12 for the Wightman function. Similarly, we use T[12] for time-ordered product given later in eq. (3.32).

about the behavior at $p^2 = 0$ ($w_1, w_2 \to -\infty$): it implies that the highest-spin vertex function is a linear combination of the first two functions in the list (3.20),

$$V_{\Delta,\ell,\ell}^{\mathrm{T}[12]}(w_1, w_2) = (-w_2)^{(\Delta_1 + \Delta_2 - \Delta - \ell - d)/2} \left[(1 - w_1 - w_2)^2 - 4 w_1 w_2\right]^{\ell/2}$$
$$\times \left[A \left(\frac{w_1}{w_2}\right)^{\Delta_1 - d/2} F_{\Delta_1 \Delta; \Delta_2, \ell}\left(\frac{w_1}{w_2}, \frac{1}{w_2}\right) + B\, F_{\widetilde{\Delta}_1 \Delta; \Delta_2, \ell}\left(\frac{w_1}{w_2}, \frac{1}{w_2}\right)\right]. \quad (3.28)$$

The coefficients $A$ and $B$ can be fixed by symmetry arguments and by the relation to the Wightman function. The singularities of the time-ordered product have been analyzed in ref. [69], where it was found that there is a branch-point singularity at $w_1 = 0$ when $\Delta_1 < \frac{d}{2}$, and that the coefficient of this singularity is related to the Wightman function by a multiplicative factor, giving

$$\lim_{w_1 \to 0} (-w_1 - i\epsilon)^{d/2 - \Delta_1} V_{\Delta,\ell,\ell}^{\mathrm{T}[12]}(w_1, w_2) = \frac{1}{2i \sin\left[\pi\left(\frac{d}{2} - \Delta_1\right)\right]} \lim_{w_1 \to 0_+} w_1^{d/2 - \Delta_1} V_{\Delta,\ell,\ell}^{12}(w_1, w_2). \quad (3.29)$$

This implies

$$A = \frac{i^{\ell - 1} (4\pi)^{d+1} \Gamma\left(\frac{d}{2} - \Delta_1\right)(\Delta - 1)_\ell}{2^{\Delta_1 + \Delta_2 + \Delta + \ell + 1} \Gamma\left(\Delta - \frac{d}{2} + 1\right) \Gamma\left(\frac{\Delta_2 - \Delta_1 + \Delta + \ell}{2}\right) \Gamma\left(\frac{\Delta_1 + \Delta_2 - \Delta + \ell}{2}\right)}. \quad (3.30)$$

The coefficient $B$ can be fixed in turn by imposing the exchange symmetry between the operators $\phi_1$ and $\phi_2$, which is built into the time-ordered product but does not appear obvious in eq. (3.28). This constraint can be resolved by going from the basis of Appell $F_4$ functions (3.20) to the symmetric basis (3.17). We find that the result is symmetric under the simultaneous exchange $w_1 \leftrightarrow w_2$ and $\Delta_1 \leftrightarrow \Delta_2$ if and only if

$$B = \frac{\Gamma\left(\Delta_1 - \frac{d}{2}\right) \Gamma\left(\frac{\Delta_2 - \Delta_1 + \Delta + \ell}{2}\right) \Gamma\left(\frac{\Delta - \Delta_1 - \Delta_2 + \ell + d}{2}\right)}{\Gamma\left(\frac{d}{2} - \Delta_1\right) \Gamma\left(\frac{\Delta_1 - \Delta_2 + \Delta + \ell}{2}\right) \Gamma\left(\frac{\Delta_1 + \Delta_2 + \Delta + \ell - d}{2}\right)} A, \quad (3.31)$$

after which we arrive at

$$V_{\Delta,\ell,\ell}^{\mathrm{T}[12]}(w_1, w_2) = \frac{i^{\ell-1}(4\pi)^{d+1} 2^{-(\Delta_1 + \Delta_2 + \Delta + \ell + 1)}(\Delta - 1)_\ell \left[(1 - w_1 - w_2)^2 - 4 w_1 w_2\right]^{\ell/2}}{\Gamma\left(\frac{\Delta_1 - \Delta_2 + \Delta + \ell}{2}\right) \Gamma\left(\frac{\Delta_2 - \Delta_1 + \Delta + \ell}{2}\right) \Gamma\left(\frac{\Delta_1 + \Delta_2 - \Delta + \ell}{2}\right) \Gamma\left(\frac{\Delta_1 + \Delta_2 + \Delta + \ell - d}{2}\right)}$$
$$\times \Big[ f_{\Delta_1 \Delta_2; \Delta, \ell} (-w_1)^{\Delta_1 - d/2} (-w_2)^{\Delta_2 - d/2} F_{\Delta_1 \Delta_2; \Delta, \ell}(w_1, w_2)$$
$$+ f_{\Delta_1 \widetilde{\Delta}_2; \Delta, \ell} (-w_1)^{\Delta_1 - d/2} F_{\Delta_1 \widetilde{\Delta}_2; \Delta, \ell}(w_1, w_2)$$
$$+ f_{\widetilde{\Delta}_1 \Delta_2; \Delta, \ell} (-w_2)^{\Delta_2 - d/2} F_{\widetilde{\Delta}_1 \Delta_2; \Delta, \ell}(w_1, w_2)$$
$$+ f_{\widetilde{\Delta}_1 \widetilde{\Delta}_2; \Delta, \ell} F_{\widetilde{\Delta}_1 \widetilde{\Delta}_2; \Delta, \ell}(w_1, w_2) \Big], \quad (3.32)$$

where we have defined

$$f_{\Delta_a \Delta_b; \Delta, \ell} = \frac{\Gamma\left(\frac{d}{2} - \Delta_a\right) \Gamma\left(\frac{d}{2} - \Delta_b\right) \Gamma\left(\frac{\Delta_a + \Delta_b + \Delta + \ell - d}{2}\right)}{\Gamma\left(1 - \frac{\Delta_a + \Delta_b - \Delta + \ell}{2}\right)}. \quad (3.33)$$

This result is analytic in the scaling dimensions $\Delta_1, \Delta_2$, with the exception of poles at $\Delta_1, \Delta_2 = \frac{d}{2} + n$ with integer $n$. This means that the result (3.32) is valid in all generality,

and not only in the regime $\Delta_1 < \frac{d}{2}$ used to fix the coefficient $A$. The apparent divergences when $\Delta_1, \Delta_2 = \frac{d}{2} + n$ are reminiscent of the anomalies discussed in refs. [70,71]: they appear precisely when the powers of $w_1$ and $w_2$ are integer, in which case the terms of the different hypergeometric series can be mixed. Resolving these special cases by analytic continuation in $\Delta_i$ (or in $d$), one finds cancellations between the coefficients $f_{\Delta_a \Delta_b; \Delta, \ell}$, so that the vertex functions remain in fact finite, however with the appearance of logarithms of $w_1$ and $w_2$. An example realizing the case $\Delta_1 = \Delta_2 = \frac{d}{2}$ is presented below in section 4.3.

**Other orderings** Besides the Wightman function and the time-ordered product, other orderings of the operators can be envisioned, but they can all be related to the two results (3.27) and (3.32): ordinary commutators are nothing but the difference between two Wightman functions; causal commutators (i.e. advanced or retarded) are similarly related to the sum or difference between the time-ordered product and the Wightman function, and as such they coincide with the former when both momenta $p_1$ and $p_2$ are space-like; the anti-time-ordered product is complex conjugated to the time-ordered one. In order to obtain a complete picture of all the possible vertex functions, one needs of course to go beyond the kinematics assumptions made in deriving eqs. (3.27) and (3.32). Causal commutators would be a useful tool to do so, as they have well-understood domains of analyticity in terms of complex momenta. We leave this task for future work.

One important property that is shared by all orderings is the vanishing of the highest-spin vertex function when the intermediate operator takes a double-trace dimension, i.e. when $\Delta = \Delta_1 + \Delta_2 + \ell + 2n$ with integer $n$. This ensures the triviality of Gaussian theories in momentum space, and places on equal footing the contributions of single- and double-trace operators in large-$N$ theories. It also implies that the conformal partial wave expansion (1.2) shares the "dispersive" property of the double discontinuity in position space [38].

## 3.3 Lower spin functions

With the knowledge of the highest-spin vertex function, the computation of all remaining vertex functions $V^{[12]}_{\Delta,\ell,m}$ with $m < \ell$ does not require solving more partial differential equations, nor making assumptions about boundary conditions, as eq. (3.8) gives a simple two-step algebraic recursion relation: since the tensors $\varepsilon'_m$ are orthogonal to each other, we get

$$
\begin{aligned}
V^{[12]}_{\Delta,\ell,m}(w_1,w_2) = {} & \frac{m+1}{(\ell-m)(\Delta+m-1)} \\
& \times \Bigg[ \frac{1}{\sqrt{(1-w_1-w_2)^2 - 4w_1 w_2}} \widehat{D}_0 V^{[12]}_{\Delta,\ell,m+1}(w_1,w_2) \\
& \quad + \frac{(m+2)(\Delta-m-d)(d-1+\ell+m)}{(d-1+2m)(d+1+2m)} V^{[12]}_{\Delta,\ell,m+2}(w_1,w_2) \Bigg],
\end{aligned}
\tag{3.34}
$$

using the first-order differential operator $\widehat{D}_0$ defined in eq. (3.13). The last line must be dropped in the first step of the recursion, namely when $m = \ell - 1$; equivalently, the recursion relation can be defined in terms of the functions $V^{[12]}_{\Delta,\ell,\ell}(w_1,w_2)$ computed above and of $V^{[12]}_{\Delta,\ell,\ell+1}(w_1,w_2) = 0$. Note that the operator $\widehat{D}_0$ is odd under the permutation $1 \leftrightarrow 2$, which is consistent with the fact that the Gegenbauer polynomials $C^{(d-3)/2}_m(\cos\theta)$ is similarly odd under $\cos\theta \to -\cos\theta$ for odd $m$: in this way the exchange symmetry $1 \leftrightarrow 2$ of the time-ordered product is preserved for all $m$.

Derivatives of Appell $F_4$ functions can be again expressed in terms of linear combinations of $F_4$ functions, so one could in principle write down closed-form formulae for the vertex functions $V^{[12]}_{\Delta,\ell,m}(w_1,w_2)$ with the different operator orderings. This exercise is not particularly

enlightening, though, and the recursion relation (3.34) seems simple enough to provide the most efficient implementation of the vertex functions. In fact, since $\widehat{D}_0$ is a first-order differential operator that only raises powers of $w_1$ and $w_2$, the vertex functions with $m < \ell$ inherit the analytic structure of the highest-spin function.

The case in which the operator $\mathcal{O}^{\mu_1\cdots\mu_\ell}$ is a conserved current deserves special attention. It turns out that the differential operator $\widehat{D}_0$ annihilates $V^{[12]}_{\Delta,\ell,\ell}(w_1,w_2)$ whenever the unitarity bound is saturated ($\Delta = d - 2 + \ell$) and the two operators $\phi_1$ and $\phi_2$ have identical scaling dimensions ($\Delta_1 = \Delta_2$), two conditions that must be met for $\mathcal{O}$ to be conserved. This implies that $V^{[12]}_{\Delta,\ell,\ell-1}(w_1,w_2) = 0$. Moreover, the factor $(\Delta - m - d)$ in eq. (3.34) ensures that the second step of the recursion is trivial as well, and so $V^{[12]}_{\Delta,\ell,\ell-2}(w_1,w_2) = 0$. It follows immediately that all vertex functions with $m < \ell$ vanish in this case. This is not a surprise: conserved currents only admit transverse polarizations, while any longitudinal polarization defines a null state. This property is essential in the conformal partial wave expansion, as it implies that one does not need to sum over $m < \ell$ for conserved currents, and hence that the singularities in the definition (2.40) of the coefficient $C_{\Delta,\ell,m}$ never occurs.

## 3.4 Special kinematic limits

We will finally examine several special kinematic limits in which the vertex functions admits a simpler form, and compare the conformal partial waves in these limits with results already existing in the literature.

**Light-like limit** The first interesting case is the limit $w_1, w_2 \to 0$ of the vertex function with a time-ordered product. As the Appell $F_4$ function is analytic around the point $(w_1, w_2) = (0,0)$, one can see from eq. (3.32) that the limit exists if $\Delta_1, \Delta_2 > \frac{d}{2}$ (the opposite situation is discussed below). In this case, the recursion relation (3.34) becomes an algebraic equation,

$$V^{\mathrm{T}[12]}_{\Delta,\ell,m}(0,0) = \frac{m+1}{(\ell-m)(\Delta+m-1)}\Bigg[(\Delta_1 - \Delta_2)V^{[12]}_{\Delta,\ell,m+1}(0,0) \\ + \frac{(m+2)(\Delta-m-d)(d-1+\ell+m)}{(d-1+2m)(d+1+2m)}\,V^{\mathrm{T}[12]}_{\Delta,\ell,m+2}(0,0)\Bigg], \quad (3.35)$$

which is solved by

$$V^{\mathrm{T}[12]}_{\Delta,\ell,m}(0,0) = V^{\mathrm{T}[12]}_{\Delta,\ell,\ell}(0,0)\frac{(-2)^{\ell-m}\ell!}{m!(\ell-m)!}\frac{\left(\frac{\Delta_2-\Delta_1+\Delta-\ell-d+2}{2}\right)_{\ell-m}}{(\Delta-1+m)_{\ell-m}} \\ \times {}_3F_2\left(\begin{array}{c} -(\ell-m),\ d-\Delta-1+m,\ \frac{d-2+2m}{2} \\ \frac{\Delta_1-\Delta_2+d-\Delta-\ell+2m}{2},\ d-2+2m \end{array};1\right). \quad (3.36)$$

Remarkably, the recursion relation only depends on the difference between the external scaling dimensions, $\Delta_1 - \Delta_2$, and not on their sum. In fact, when $\Delta_1 = \Delta_2 (\equiv \Delta_\phi)$, all vertex functions with odd $\ell - m$ vanish identically, and one obtains

$$V^{\mathrm{T}[\phi\phi]}_{\Delta,\ell,\ell-2n}(0,0) = \frac{(-1)^n\ell!}{2^{2n}n!(\ell-2n)!}\frac{\left(\frac{\Delta-\ell-d+2}{2}\right)_n}{\left(\frac{d-1}{2}+\ell-2n\right)_n\left(\frac{3-\Delta-\ell}{2}\right)_n}V^{\mathrm{T}[\phi\phi]}_{\Delta,\ell,\ell}(0,0). \quad (3.37)$$

The conformal partial waves (1.4) can then be written as

$$G_{\Delta,\ell}(w_i = 0, \cos\theta) = \frac{2^{2\Delta+1}\Gamma\left(\Delta-\frac{d}{2}+1\right)\Gamma(\Delta+\ell)}{(4\pi)^{(d+2)/2}(\Delta-1)_\ell}\left[V^{[\phi\phi]}_{\Delta,\ell,\ell}(0,0)\right]^2 g_{\Delta,\ell}(\cos\theta), \quad (3.38)$$

where $g_{\Delta,\ell}$ is a polynomial of degree $\ell$ in $\cos\theta$, defined by

$$g_{\Delta,\ell}(\cos\theta) = \sum_{n=0}^{\ell/2} \frac{(-1)^n \ell!(2n)!}{2^{\ell+2n+1}(n!)^2} \frac{d-3+2\ell-4n}{\left(2-\frac{d}{2}-\ell\right)_n \left(\frac{d-3}{2}\right)_{\ell-n+1}} \frac{\left(\frac{2-\Delta-\ell}{2}\right)_n \left(\frac{2-\tilde{\Delta}-\ell}{2}\right)_n}{\left(\frac{3-\Delta-\ell}{2}\right)_n \left(\frac{3-\tilde{\Delta}-\ell}{2}\right)_n} \mathcal{C}_{\ell-2n}^{(d-3)/2}(\cos\theta).$$

(3.39)

These are precisely the polynomials defined in ref. [63], where the light-like limit of the conformal partial wave expansion was studied. The $g_{\Delta,\ell}$ have the remarkable property of interpolating continuously between the Gegenbauer polynomials $\mathcal{C}_\ell^{(d-3)/2}(\cos\theta)$ when $\Delta - \ell$ saturates the unitarity bound, and $\mathcal{C}_\ell^{(d-2)/2}(\cos\theta)$ when $\Delta - \ell \to \infty$. Using

$$V_{\Delta,\ell,\ell}^{\mathrm{T}[\phi\phi]}(0,0) = \frac{i^{\ell-1}(4\pi)^{d+1}2^{-(2\Delta_\phi+\Delta+\ell+1)}\Gamma\left(\Delta_\phi - \frac{d}{2}\right)^2 \Gamma\left(\frac{\Delta+\ell+d}{2} - \Delta_\phi\right)(\Delta-1)_\ell}{\Gamma\left(\frac{\Delta+\ell}{2}\right)^2 \Gamma\left(\Delta_\phi - \frac{\Delta-\ell}{2}\right)\Gamma\left(\Delta_\phi + \frac{\Delta+\ell-d}{2}\right)\Gamma\left(\Delta_\phi + \frac{\Delta-\ell}{2} - d+1\right)},$$

(3.40)

one can also verify that the multiplicative coefficient in eq. (3.38) also matches the definition of $G_{\Delta,\ell}(\cos\theta)$ in ref. [63].[6]

**Scattering amplitude and form factor** An alternative, maybe more interesting situation is the asymptotic limit $w_i \to 0$ in the case $\Delta_i < \frac{d}{2}$. The time-ordered vertex function (3.32) diverges in this case, but one obtains a well-defined limit by multiplying it with the appropriate powers of $w_i$. The peculiarity of this limit is that it allows to define objects that have an interesting field-theoretical interpretation: consider

$$F(s,t,u) \equiv \left[\prod_{i=1}^{3} \lim_{p_i^2 \to 0_-} (-p_i^2)^{d/2-\Delta_i}\right] \langle\!\langle \mathrm{T}[\phi_1(p_1)\phi_2(p_2)\phi_3(p_3)\phi_4(-p_1-p_2-p_3)]\rangle\!\rangle, \quad (3.41)$$

as well as the subsequent limit $p_4^2 = s+t+u \to 0$,

$$A(s,t) \equiv \lim_{p_4^2 \to 0_-} (-p_4^2)^{d/2-\Delta_4} F(s,t,u). \quad (3.42)$$

$F$ and $A$ have been interpreted respectively as a "form factor" and as an "amplitude" [69]: on the one hand, they are related to the time-ordered correlation function in eq. (3.41) by a procedure similar to the LSZ reduction in massive quantum field theory; on the other hand, from a careful analysis of the singularities in the Fourier transform from position space, one can show that they admit a conformal partial wave expansion that is reminiscent of the "unitarity cuts" method relating higher-point amplitudes to lower-point ones. This latter property follows from the fact that the form factor is related to the same limit of the partially-time-ordered function,

$$F(s,t,u) \sim \left[\prod_{i=1}^{3} \lim_{p_i^2 \to 0_-} (-p_i^2)^{d/2-\Delta_\phi}\right] \langle\!\langle \mathrm{T}[\phi_1(p_1)\phi_2(p_2)]\,\mathrm{T}[\phi_3(p_3)\phi_4(-p_1-p_2-p_3)]\rangle\!\rangle, \quad (3.43)$$

provided that the limit is taken with $p_1$, $p_2$ backward-directed, and $p_3$ forward-directed. The equivalence is up to a phase involving the dilatation operator (i.e. it depends on the scaling dimension $\Delta$ of the intermediate operator in the conformal partial wave). We do not want to re-derive here the results of ref. [69], but we can directly make use of the right-hand side of the equivalence (3.43) to provide a simple computation of the form factor $F$ and of the amplitude $A$ using our conformal partial waves, up to the aforementioned phase.

---

[6]Eq. (1.14) in ref. [63] contains a typo: the numerator is missing a factor of $\Gamma(\Delta + \ell)$ (corrected in the arXiv version).

The computation involves two distinct vertex functions, $V^{\mathrm{T}[12]}_{\Delta,\ell,m}$ and the complex conjugate of $V^{\bar{\mathrm{T}}[34]}_{\Delta,\ell,m}$. $V^{\mathrm{T}[12]}_{\Delta,\ell,m}$ is evaluated with both $w_1$ and $w_2 \to 0$. In this case the computation proceeds as in the light-like limit discussed above: the recursion relation (3.34) for the vertex function is again algebraic, though it differs from eq. (3.35) by the replacement $\Delta_{1,2} \to d - \Delta_{1,2}$. One arrives at

$$
\lim_{w_1,w_2 \to 0_-} (-w_1)^{d/2-\Delta_1}(-w_2)^{d/2-\Delta_2} V^{\mathrm{T}[12]}_{\Delta,\ell,m}(w_1,w_2)
$$

$$
= \frac{i^{\ell-1}(4\pi)^d \Gamma\left(\frac{d}{2}-\Delta_1\right)\Gamma\left(\frac{d}{2}-\Delta_2\right)(\Delta-1)_\ell \left(\frac{\Delta_1-\Delta_2+\Delta-\ell-d+2}{2}\right)_{\ell-m} \sin\left[\pi \frac{\Delta_1+\Delta_2-\Delta+\ell}{2}\right]}{2^{\Delta_1+\Delta_2+\Delta+\ell-1}\Gamma\left(\frac{\Delta_1-\Delta_2+\Delta+\ell}{2}\right)\Gamma\left(\frac{\Delta_2-\Delta_1+\Delta+\ell}{2}\right)(\Delta-1+m)_{\ell-m}}
$$

$$
\times \frac{(-2)^{\ell-m}\ell!}{m!(\ell-m)!}{}_3F_2\left(\begin{array}{c} -(\ell-m),\ d-\Delta-1+m,\ \frac{d-2+2m}{2} \\ \frac{\Delta_2-\Delta_1+d-\Delta-\ell+2m}{2},\ d-2+2m \end{array}; 1\right). \tag{3.44}
$$

For $V^{\bar{\mathrm{T}}[34]}_{\Delta,\ell,m}$, consider first the highest-spin vertex function as defined in eq. (3.28), which in the limit $w_3 \to 0$ becomes

$$
\lim_{w_3 \to 0_-} (-w_3)^{d/2-\Delta_3} V^{\bar{\mathrm{T}}[34]}_{\Delta,\ell,\ell}(w_3,w_4)
$$

$$
= A^*(-w_4)^{(\Delta_3+\Delta_4-\Delta+\ell-2)/2}(1-w_4)^{1-\Delta_3}
$$

$$
\times {}_2F_1\left(\frac{\Delta-\ell-\Delta_3-\Delta_4+2}{2},\ \frac{\Delta_4-\Delta_3+\Delta-\ell-d+2}{2}; \Delta-\frac{d}{2}+1; \frac{1}{w_4}\right), \tag{3.45}
$$

where $A^*$ is the complex conjugate of the coefficient $A$ given in eq. (3.30), upon replacement $1 \to 3$ and $2 \to 4$. Applying the recursion relation (3.34) leads to

$$
\lim_{w_3 \to 0_-} (-w_3)^{d/2-\Delta_3} V^{\mathrm{T}[12]}_{\Delta,\ell,m}(w_1,w_2) = A^*(-w_4)^{(\Delta_3+\Delta_4-\Delta+\ell-2)/2}(1-w_4)^{1-\Delta_3} v_{\Delta,\ell,m}\left(\frac{1}{w_4}\right), \tag{3.46}
$$

where we have defined

$$
v_{\Delta,\ell,m}(z) = \sum_{j=0}^{\ell-m} \frac{2^{\ell-m}\ell!}{m!j!(\ell-m-j)!}\frac{\left(\frac{d}{2}-1+m\right)_{\ell-m-j}(\Delta-\ell-d+2+j)_{\ell-m-j}}{(d-2+2m)_{\ell-m-j}}
$$

$$
\times \frac{\left(\frac{\Delta-\ell-\Delta_3-\Delta_4+2}{2}\right)_j \left(\frac{\Delta_4-\Delta_3+\Delta-\ell-d+2}{2}\right)_j}{(2-\Delta-\ell)_{\ell-m}\left(\Delta-\frac{d}{2}+1\right)_j}
$$

$$
\times z^j {}_2F_1\left(\frac{\Delta-\ell-\Delta_3-\Delta_4+2}{2}+j,\ \frac{\Delta_4-\Delta_3+\Delta-\ell-d+2}{2}+j; \Delta-\frac{d}{2}+1+j; z\right). \tag{3.47}
$$

The conformal partial waves corresponding to the form factor follow simply from plugging the vertex functions (3.44) and (3.46) in eq. (1.4). This gives a closed-form expression for distinct $\Delta_i$ that was lacking in ref. [69], as well as a new representation of the result in the case of identical operators ($\Delta_i = \Delta_\phi$): the function $f_{\Delta,\ell}(z,\cos\theta)$ computed in appendix B of ref. [69] using a method based on the conformal Casimir equations can be written

$$
f_{\Delta,\ell}(z,\cos\theta) = z^{(\Delta-\ell-2\Delta_\phi)/2}(1-z)^{1-\Delta_\phi}
$$

$$
\times \sum_{n=0}^{\ell/2} \frac{(-1)^n \ell!}{2^\ell n!}\frac{\left(\frac{d-3}{2}+\ell-2n\right)\left(\frac{\Delta-\ell-d+2}{2}\right)_n}{\left(\frac{d-3}{2}\right)_{\ell-n+1}\left(\frac{3-\Delta-\ell}{2}\right)_n}\mathcal{C}^{(d-3)/2}_{\ell-2n}(\cos\theta)v_{\Delta,\ell,\ell-2n}(z), \tag{3.48}
$$

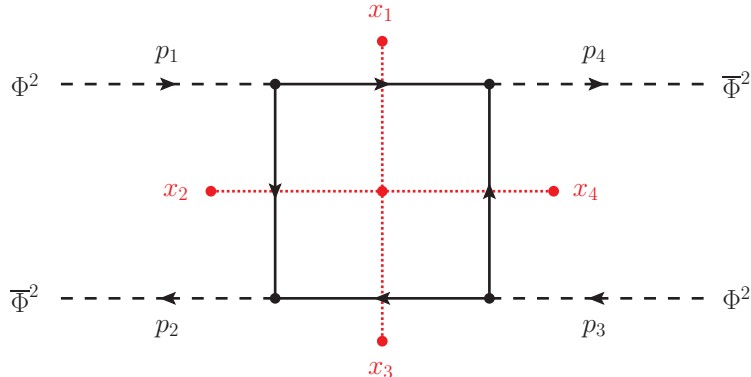

Figure 3: The scalar box diagram corresponding to the 4-point function (4.1) in the theory of a free complex scalar field. The solid lines are propagators of the field $\Phi$, with arrows indicating the charge flow. The external, dashed lines correspond to the composite operators $\Phi^2$ and $\overline{\Phi}^2$. The dual graph in terms of the variables $x_i - x_{i+1} = p_i$ is shown in red, with dotted lines.

in terms of $z = 1/w_4 = p_4^2/s$. Similarly, for the scattering amplitude (3.30) one recovers the polynomials $g_{\Delta,\ell}$ of eq. (3.39) in the case of identical external operators, as well as a new set of polynomials obtained by plugging the vertex function (3.44) twice in eq. (1.4) when the external operators are distinct.

These examples of special kinematics are far from being exhaustive, but they illustrate how our results can be efficiently used in various specific configurations. This concludes the derivation of the vertex functions, and we move next to a complete physical example where the conformal partial wave expansion are put to good use.

# 4 Example: the scalar box integral in 4 dimensions

We present in this section an example in which the conformal partial wave expansion can be explicitly compared with a known 4-point function. We take the theory of a free, complex scalar field $\Phi$, and consider the time-ordered 4-point function involving the composite operator $\Phi^2$ and its complex conjugate $\overline{\Phi}^2$,

$$\langle 0| \, T[\Phi^2(p_1)\overline{\Phi}^2(p_2)\Phi^2(p_3)\overline{\Phi}^2(p_4)] \, |0\rangle \, . \tag{4.1}$$

This correlation function is given in terms of a single Feynman diagram shown in figure 3. In $d = 4$ dimensions, the integral over the loop momentum can be performed in arbitrary kinematics, and the result is a Bloch-Wigner dilogarithm [72–74]. The simplicity of this result can be traced back to presence of a dual conformal symmetry at the level of the integrand [45, 46].

## 4.1 CFT optical theorem

The 4-point function (4.1) in itself does not admit a decomposition into conformal partial waves: the time-ordered product runs over all 4 operators, and as such it does not respect the structure of the correlator (1.1). However, the real part of that function does, by an identity that is sometimes referred to as the "CFT optical theorem" [31, 32] (see also ref. [75]): by a combinatoric identity involving the time-ordered product and its hermitian conjugate $\overline{T}$, it can

be shown that

$$\langle 0|\,\mathrm{T}[\Phi^2(p_1)\overline{\Phi}^2(p_2)\Phi^2(p_3)\overline{\Phi}^2(p_4)]\,|0\rangle + \langle 0|\,\overline{\mathrm{T}}[\Phi^2(p_1)\overline{\Phi}^2(p_2)\Phi^2(p_3)\overline{\Phi}^2(p_4)]\,|0\rangle$$

$$= \langle 0|\,\mathrm{T}[\Phi^2(p_1)\overline{\Phi}^2(p_2)]\,\overline{\mathrm{T}}[\Phi^2(p_3)\overline{\Phi}^2(p_4)]\,|0\rangle\,, \qquad (4.2)$$

provided that the following conditions on the momenta and Mandelstam invariants are fulfilled:

$$p_i^2 < 0, \qquad\qquad s > 0, \qquad\qquad t, u < 0\,. \qquad (4.3)$$

The right-hand side of this optical theorem is precisely in the form of eq. (1.1), and we can therefore define

$$G(s, w_i, \cos\theta) = \langle\!\langle\mathrm{T}[\Phi^2(p_1)\overline{\Phi}^2(p_2)]\,\overline{\mathrm{T}}[\Phi^2(p_3)\overline{\Phi}^2(p_4)]\rangle\!\rangle\,. \qquad (4.4)$$

At the same time, the left-hand side of eq. (4.2) corresponds to (twice) the real part of the scalar box integral. This can be compared with the known results that give [45, 74][7]

$$G(s, w_i, \cos\theta) = \frac{(2\pi)^7}{su}\frac{\log\left(\frac{1-1/z}{1-1/\bar{z}}\right)}{z-\bar{z}}\,. \qquad (4.5)$$

$z$ and $\bar{z}$ are related to the dual conformal cross ratios, given in terms of $p_i = x_i - x_{i+1}$ by the two equations

$$z\bar{z} = \frac{x_{12}^2 x_{34}^2}{x_{13}^2 x_{24}^2} = \frac{p_1^2 p_3^2}{su}, \qquad\qquad (1-z)(1-\bar{z}) = \frac{x_{14}^2 x_{23}^2}{x_{13}^2 x_{24}^2} = \frac{p_2^2 p_4^2}{su}\,. \qquad (4.6)$$

## 4.2 IR divergences

In spite of the relative simplicity of (4.5) in terms of $z$ and $\bar{z}$, the dependence on $w_i$ and $\cos\theta$ is quite intricate: determining $z$ and $\bar{z}$ requires solving a quadratic equation involving the Mandelstam invariant $u$, which is itself given by

$$u = -\frac{s}{2}\Big[\, 1 - w_1 - w_2 - w_3 - w_4 - (w_1 - w_2)(w_3 - w_4)$$

$$- \cos\theta\sqrt{(1-w_1-w_2)^2 - 4w_1 w_2}\sqrt{(1-w_3-w_4)^2 - 4w_3 w_4}\,\Big]\,. \qquad (4.7)$$

A comparison with the conformal partial wave expansion at generic kinematics is quite difficult to achieve. Instead, we will focus on the IR divergences of this box integral when some or all of the $p_i^2$ approach zero. In the simplest case where all "masses" are identical ($w_i = w$), we have

$$z, \bar{z} = \frac{1}{2}\left(1 \pm \sqrt{1 + \frac{8w^2}{(1-4w)(1-\cos\theta)}}\,\right), \qquad (4.8)$$

and, as $w \to 0$,

$$G(s, w_i = w, \cos\theta) = -\frac{512\pi^7}{s^2(1-\cos\theta)}\log\left(\frac{2w^2}{1-\cos\theta}\right) + \dots \qquad (4.9)$$

The real part of the box integral has a logarithmic divergence in $w$, and moreover a pole in the forward limit $\cos\theta \to 1$.

---

[7] Our real part of the box integral differs from the usual conventions by a multiplicative factor of $2^{10}\pi^8$ that has to do with the use of the standard CFT normalization of operators in position space: on the one hand, the free field propagator in 4 dimensions is $\langle\!\langle\mathrm{T}[\Phi(p)\overline{\Phi}(-p)]\rangle\!\rangle = 4\pi^2 i/p^2$, accounting for an overall factor of $(4\pi^2)^4$; on the other hand, each vertex involving the properly-normalized CFT operator $\frac{1}{\sqrt{2}}\Phi^2$ gives rise to a factor of $\sqrt{2}$, contributing in total another factor of 4 to our box integral.

To perform the comparison with the conformal partial waves, one would like to expand $G$ in spin partial waves first, and write

$$G(s, w_i, \cos\theta) = s^{-2} \sum_{\ell=0}^{\infty} \left(\ell + \tfrac{1}{2}\right) a_\ell(w_i) P_\ell(\cos\theta), \qquad (4.10)$$

or equivalently, using the orthogonality of Legendre polynomials,

$$a_\ell(w_i) = \int_{-1}^{1} d\cos\theta \, s^2 G(s, w_i, \cos\theta). \qquad (4.11)$$

Note that the asymptotic limit in eq. (4.9) does not commute with the partial wave expansion, since the pole in $\cos\theta$ is not integrable. One must work instead with generic $w$, and take the limit $w \to 0$ at the end. To do so, note that $G$ can be rewritten as

$$G(s, w_i, \cos\theta) = -\frac{64\pi^7}{s^2 \Omega_{12} \Omega_{34}} \frac{\partial}{\partial \cos\theta} \left[ \log\left(\frac{1 - 1/z}{1 - 1/\bar{z}}\right) \right]^2. \qquad (4.12)$$

The angular integral can now be performed using integration by parts after taking the IR limit in the logarithm. For identical $w_i = w$, the limit $w \to 0$ gives

$$G(s, w_i = w, \cos\theta) = -\frac{256\pi^7}{s^2} \frac{\partial}{\partial \cos\theta} \left[ \log\left(\frac{2w^2}{1 - \cos\theta}\right) + \ldots \right]^2, \qquad (4.13)$$

and thus

$$a_\ell(w_i = w) = 1024\pi^7 [\log(-w) + H_\ell]^2 + \ldots, \qquad (4.14)$$

where $H_\ell$ is the $\ell$-th harmonic number, and the ellipsis indicates that we have neglected terms of order $w \log(-w)$ and higher. For each individual partial wave the leading IR divergence is now a double logarithm in $w$.

To study a slightly more general form, one can also consider the situation in which the masses are equal two-by-two, say $w_4 = w_1$ and $w_3 = w_2$, and take the limit $w_1 \to 0$ while keeping $w_2$ generic. Using eq. (4.12), one arrives at

$$a_\ell(w_i) = \frac{256\pi^7}{(1 - w_2)^2} \left[ \log(-w_1) + \log(-w_2) - 2\log(1 - w_2) + 2H_\ell \right]^2 + \ldots, \qquad (4.15)$$

neglecting terms that vanish as $w_1 \to 0$. We shall now see in the next section how the partial waves (4.14) and (4.15) are matched one-to-one by the conformal partial waves.

## 4.3 Conformal partial wave expansion

To compute the conformal partial waves, one simply needs to apply the recipe given in sections 2 and 3 to the 4-point function (4.4). We work with $d = 4$, and with scaling dimensions of the external operator that are all equal, $\Delta_i = 2$. Since the theory is non-interacting, it is straightforward to determine the spectrum of primary operators that enter the OPE of $\Phi^2$ and $\bar{\Phi}^2$, as these must be composites of the free field. There are two families of such operators:

- Operators of the schematic form $[\bar{\Phi}^2 \Box^n \partial^\ell \Phi^2]$, that have spin $\ell$ and scaling dimensions $\Delta = 4 + \ell + 2n$. As these are precisely double-trace dimensions, the vertex functions involving them vanish. These operators do not contribute to the conformal partial wave expansion in momentum space.

- Operators of the form $[\overline{\Phi}\partial^\ell\Phi]$, with spin $\ell$ and scaling dimensions $\Delta = 2+\ell$. There is one such operator for every spin $\ell \geq 0$, and each of them is a conserved current saturating the unitarity bound, with the exception of the scalar operator $[\overline{\Phi}\Phi]$.

Therefore, the conformal partial wave decomposition (1.2) is a sum over integers labeled by the spin $\ell$, with each term given by single polarization $m = \ell$. Plugging the operator data in the coefficient $C_{\Delta,\ell,\ell}$ of eq. (2.40), one gets

$$G(s, w_i, \cos\theta) = s^{-2}\sum_{\ell=0}^{\infty}\left(\ell + \tfrac{1}{2}\right)P_\ell(\cos\theta)a_\ell(w_i), \tag{4.16}$$

where now

$$a_\ell(w_i) = \lambda^2_{\Phi^2\overline{\Phi}^2[\overline{\Phi}\partial^\ell\Phi]}\frac{2^{3\ell}(\ell!)^4}{\pi^3(2\ell)!}V^{\mathrm{T}[\phi\phi]}_{2+\ell,\ell,\ell}(w_1,w_2)V^{\mathrm{T}[\phi\phi]}_{2+\ell,\ell,\ell}(w_3,w_4)^*. \tag{4.17}$$

The computation of the vertex functions is slightly more complicated, as one cannot simply plug the relevant scaling dimensions in eq. (3.32): the coefficients $f_{\Delta_1\Delta_2;\Delta,\ell}$ diverge precisely when $\Delta_i = \frac{d}{2}$. Fortunately, it is straightforward to regularize the computation of these vertex functions by analytic continuation in $d$: the free scalar theory can be defined in any $d \neq 4$, and we shall see that the limit $d \to 4$ ends up to be finite, due to cancellations among the different terms in eq. (3.32).

In generic $d$, the scaling dimension of $\Phi^2$ and $\overline{\Phi}^2$ is $\Delta_i = d-2$, and the conserved currents entering the OPE have $\Delta = d - 2 + \ell$. The coefficients $f_{\Delta_1\Delta_2;\Delta,\ell}$ appearing in eq. (3.32) take the values

$$f_{\Delta_1\Delta_2;\Delta,\ell} = \Gamma(d-3+\ell)\Gamma\left(\tfrac{4-d}{2}\right), \quad f_{\widetilde{\Delta}_1\widetilde{\Delta}_2;\Delta,\ell} = \ell!\,\Gamma\left(\tfrac{d-4}{2}\right), \quad f_{\Delta_1\widetilde{\Delta}_2;\Delta,\ell} = f_{\widetilde{\Delta}_1\Delta_2;\Delta,\ell} = 0. \tag{4.18}$$

Since the combination $f_{\Delta_1\Delta_2;\Delta,\ell} + f_{\widetilde{\Delta}_1\widetilde{\Delta}_2;\Delta,\ell}$ vanishes in the limit $d \to 4$, the individual divergences cancel out and the powers $(-w_i)^{\Delta_i-d/2}$ turn into logarithms. The general structure of the vertex function is still quite complicated, as it involves Appell $F_4$ functions and derivatives of them with respect to their parameters. But it is simple to obtain the limit in which one of their argument $w_i$ is small. For instance, as $w_1 \to 0$, we get

$$V^{\mathrm{T}[\phi\phi]}_{2+\ell,\ell,\ell}(w_1,w_2) = 8\pi^5 i^{\ell+1}\frac{(2\ell)!}{2^{2\ell}(\ell!)^3}\frac{1}{1-w_2}\left[\log(-w_1) + \log\left(\frac{-w_2}{(1-w_2)^2}\right) + 2H_\ell + \dots\right]. \tag{4.19}$$

This function already bears resemblance to the partial wave decomposition obtained above. Assuming all identical $w_i = w$, one gets

$$a_\ell(w_i = w) = 256\pi^7\frac{(2\ell)!}{2^\ell(\ell!)^2}\lambda^2_{\Phi^2\overline{\Phi}^2[\overline{\Phi}\partial^\ell\Phi]}\left[\log(-w) + H_\ell + \dots\right]^2, \tag{4.20}$$

while with $w_4 = w_1$ and $w_3 = w_2$ we have

$$a_\ell(w_i) = 64\pi^7\frac{(2\ell)!}{2^\ell(\ell!)^2}\lambda^2_{\Phi^2\overline{\Phi}^2[\overline{\Phi}\partial^\ell\Phi]}\left[\log(-w_1) + \log(-w_2) - 2\log(1-w_2) + 2H_\ell + \dots\right]^2. \tag{4.21}$$

These coefficients match precisely eqs. (4.14) and (4.15) provided that

$$\lambda^2_{\Phi^2\overline{\Phi}^2[\overline{\Phi}\partial^\ell\Phi]} = \frac{2^{\ell+2}(\ell!)^2}{(2\ell)!}. \tag{4.22}$$

This is indeed the known expression for the OPE coefficients in the complex scalar field theory in $d = 4$ dimensions. This example gives therefore a non-trivial verification of our general results in a specific conformal field theory.

# 5 Conclusions

In this work, we have presented a computation of the conformal partial waves in momentum space, with an emphasis on their factorization properties: they are given as a finite sum over vertex functions and polynomials in the scattering angle. The terms of this sum are in one-to-one correspondence with the ordinary spin partial waves. The vertex functions are simple to obtain using a recursion relation descending from the highest-spin function, and the latter are expressed in terms of Appell $F_4$ double hypergeometric functions, which are straightforward to evaluate numerically. In some particular cases where the scaling dimensions of the external operators take (half-)integer values, the vertex function must be obtained by analytic continuation: we gave an example where this situation happens in section 4. The outcome of this work is a new mathematical tool for the study of conformal field theory, which we hope will be useful in the future. Some directions that are certainly worth investigating are the possibility to use partial wave unitarity to put constraints on the CFT data, in the spirit of the S-matrix bootstrap [76], as well as the prospect of writing CFT dispersion relations directly in momentum space [38, 39].

We also believe that our computation of the conformal partial waves illustrates the power of Minkowski momentum-space techniques in CFT. It is well known that the orthogonality of momentum eigenstates allows to factorize the computation of correlation functions: after all, this is the reason why the majority of quantum field theory results are obtained in momentum space. What is certainly less appreciated is the structural simplicity of the operator product expansion in momentum space. While 3-point function are admittedly complicated, the fact that they depend by momentum conservation on only two momenta is very powerful: they can always be thought to live on a 2-dimensional slice of Minkowski space, and their spin can be projected onto this subspace.[8] For this reason, the dependence on the scattering angle can always be factorized in the operator product expansion in momentum space.

In fact, it is a simple exercise to generalize the computation of conformal partial waves to correlation functions involving more than four points, at least as long as one is interested in the comb channel [77–81]. Similarly, we believe that the inclusion of spinning external operators in the computation could be potentially simpler in momentum space than it is for ordinary conformal blocks in position space (see ref. [82], as well as the connection with the Mellin-space representation in refs. [83, 84]).

Finally, several properties of our conformal partial waves could (and should) be studied in the future. First and foremost, we have not touched the question of the convergence of the momentum space OPE. This convergence is actually guaranteed in a distributional sense, since CFT correlation functions in Minkowski space are tempered distributions and so are their Fourier transforms, and the position-space OPE can be shown to converge distributionally [85]. Nevertheless, the situation can be quite subtle, as illustrated by the solution to this problem in $d = 2$ dimensions [86]: depending on the kinematics, the OPE might or might not converge in a point-like manner.

Other interesting properties of the momentum-space conformal partial wave expansion have to do with the remarkable fact that our results are valid in any space-time dimension $d \geq 3$. This would allow to study them in the limit $d \to \infty$ [87, 88], or to examine whether recursion relations between various space-time dimensions can be found [89–91]. The analyticity in $d$ also implies the ability to study momentum-space correlation functions in non-integer dimensions, where unitarity is known to be broken [92, 93]. Similarly, if one projects the OPE onto ordinary spin partial waves by working at fixed $m$, then the result is not only analytic in $d$ but also in the spin $\ell$ of the exchanged operator: in this case one can imagine formulating

---

[8]As already mentioned, this also implies that the ideal framework to work with momentum-space 3-point functions is in terms of spinor variables [62].

an OPE inversion formula in momentum space [94, 95]. It would be interesting in any case to understand the concept of light-ray operators in the same context [96–98]. We leave the study of these interesting properties for future work.

## Acknowledgments

The author would like to thank the organizers of the workshop "Cosmology Meets CFT Correlators 2020" at the National Taiwan University for their hospitality, as well as the workshop's participants for numerous stimulating discussions.

# A Proof of combinatorial identities

This appendix contains proofs of several results presented in section 2. These proofs are complicated, and it seems obvious that they could be simplified by use of geometric arguments in integer dimensions. Nevertheless, an important property of these proofs is that they only use combinatorics, and as such their results are valid in any space-time dimension $d$, including non-integer $d$.

The two main results of this appendix are the proofs of the representation (2.19) for the polarization tensors, and of the decomposition (2.27) of the 2-point function. A key element is the transverse polarization tensor (2.18), and we detail its derivation first.

## A.1 Transverse polarization tensor

By assumption, we only consider tensors living in a space spanned by two orthogonal vectors $p$ and $q$. The most general tensor in this space can be written in terms of the vectors $p^\mu$, $q^\mu$, and of the invariant tensor $\eta^{\mu\nu}$.[9] We require $\varepsilon_\perp^{\mu_1\dots\mu_\ell}$ to satisfies three conditions:

(1) permutation symmetry $\varepsilon_\perp^{\mu_1\dots\mu_i\dots\mu_j\dots\mu_\ell} = \varepsilon_\perp^{\mu_1\dots\mu_j\dots\mu_i\dots\mu_\ell}$,

(2) tracelessness $\eta_{\mu_1\mu_2}\varepsilon_\perp^{\mu_1\dots\mu_\ell} = 0$, and

(3) transversality $p_{\mu_1}\varepsilon_\perp^{\mu_1\dots\mu_\ell} = 0$.

Condition (1) is simple to implement by averaging over all permutations of the Lorentz indices, which we indicate with parentheses surrounding them. Condition (2) implies that $\eta^{\mu\nu}$ and $p^\mu$ only appear in the linear combination

$$\eta_\perp^{\mu\nu} \equiv \eta^{\mu\nu} - \frac{p^\mu p^\nu}{p^2}, \tag{A.1}$$

which is the definition (2.17) of the transverse projector. We can therefore make the ansatz

$$\varepsilon_\perp^{\mu_1\dots\mu_m}(p,q) = \sum_{n=0}^{\lfloor m/2\rfloor} a_n \frac{\eta_\perp^{(\mu_1\mu_2}\cdots\eta_\perp^{\mu_{2n-1}\mu_{2n}}q^{\mu_{2n+1}}\cdots q^{\mu_m)}}{s^{m/2-n}}, \tag{A.2}$$

---

[9]The totally antisymmetric invariant tensor cannot appear in our problem, but it could appear in 2-point functions of other Lorentz representations specific to a given space-time dimension.

where the power of $s$ is chosen so as to make $\varepsilon_\perp^{\mu_1 \dots \mu_m}$ dimensionless. The coefficients $a_n$ are still to be determined. Taking the trace of this tensors gives

$$\eta_{\mu_{m-1}\mu_m} \varepsilon_\perp^{\mu_1 \dots \mu_m}(p,q) = \sum_{n=0}^{\lfloor m/2 \rfloor - 1} \frac{2(n+1)(d-5+2m-2n)a_{n+1} - (m-2n)(m-2n-1)a_n}{m(m-1)}$$
$$\times \frac{\eta_\perp^{(\mu_1 \mu_2} \cdots \eta_\perp^{\mu_{2n-1}\mu_{2n}} q^{\mu_{2n+1}} \cdots q^{\mu_{m-2})}}{s^{m/2-n-1}}. \tag{A.3}$$

By orthogonality of the terms in the sum, condition (3) implies that each of them must vanish, which gives a recursion relation for the coefficients $a_n$,

$$a_{n+1} = \frac{(m-2n)(m-2n-1)}{2(n+1)(d-5+2m-2n)} a_n, \tag{A.4}$$

solved by

$$a_n = \frac{m!}{n!(m-2n)!} \frac{1}{2^{2n}\left(\frac{d-3}{2}+m-n\right)_n} a_0. \tag{A.5}$$

Choosing $a_0 = 1$, one recovers precisely eq. (2.18). The transverse tensor $\varepsilon_\perp$ is uniquely defined up to this choice of normalization.

In order to get to eq. (2.19), one needs to contract the Lorentz indices of the transverse polarization tensors with the null vector $z$. Using $z_\mu z_\nu \eta_\perp^{\mu\nu} = -(z \cdot p)^2/s$, one gets

$$z_{\mu_1} \cdots z_{\mu_m} \varepsilon_\perp^{\mu_1 \dots \mu_m} = \sum_{n=0}^{\lfloor m/2 \rfloor} \frac{m!}{n!(m-2n)!} \frac{(-1)^n}{2^{2n}\left(\frac{d-3}{2}+m-n\right)_n} \left(\frac{z \cdot q}{\sqrt{s}}\right)^{m-2n} \left(\frac{z \cdot p}{\sqrt{s}}\right)^{2n}. \tag{A.6}$$

The sum over $n$ defines a hypergeometric series that terminates because of the factor $(m-2n)!$ in the denominator: one can equivalently write

$$z_{\mu_1} \cdots z_{\mu_m} \varepsilon_\perp^{\mu_1 \dots \mu_m} = \left(\frac{z \cdot q}{\sqrt{s}}\right)^m {}_2F_1\left(-\frac{m}{2}, -\frac{m-1}{2}; \frac{5-d}{2}-m; \frac{(z \cdot p)^2}{(z \cdot q)^2}\right). \tag{A.7}$$

Finally, we want to determine the normalization of $\varepsilon_\perp$ and compute the contraction between tensors with different reference vectors $q$ and $q'$. Using the result (2.18) once, one can write

$$\varepsilon_\perp^{\mu_1 \dots \mu_m}(p,q)\varepsilon_{\perp\mu_1\dots\mu_m}(p,q') = \sum_{n=0}^{\lfloor m/2 \rfloor} \frac{m!}{n!(m-2n)!} \frac{1}{2^{2n}\left(\frac{d-3}{2}+m-n\right)_n}$$
$$\times \frac{\eta_\perp^{\mu_1\mu_2} \cdots \eta_\perp^{\mu_{2n-1}\mu_{2n}} q^{\mu_{2n+1}} \cdots q^{\mu_m}}{s^{m/2-n}} \varepsilon_{\perp\mu_1\dots\mu_m}(p,q'). \tag{A.8}$$

Since all terms with $n > 0$ vanish by the identity $\eta_\perp^{\mu_1\mu_2} \varepsilon_{\perp\mu_1\dots\mu_m}(p,q) = 0$, this is

$$\varepsilon_\perp^{\mu_1 \dots \mu_m}(p,q)\varepsilon_{\perp\mu_1\dots\mu_m}(p,q') = \frac{q_{\mu_1} \cdots q_{\mu_m}}{s^{m/2}} \varepsilon_\perp^{\mu_1 \dots \mu_m}(p,q'). \tag{A.9}$$

Using once again the explicit result (2.18) for $\varepsilon_\perp$ on the right-hand side, one gets

$$\varepsilon_\perp^{\mu_1 \dots \mu_m}(p,q)\varepsilon_{\perp\mu_1\dots\mu_m}(p,q') = \sum_{n=0}^{\lfloor m/2 \rfloor} \frac{m!}{n!(m-2n)!} \frac{(-1)^n(-\cos\theta)^{\ell-2n}}{2^{2n}\left(\frac{d-3}{2}+m-n\right)_n}, \tag{A.10}$$

where $\theta$ is the angle between $q$ and $q'$, i.e. $q \cdot q' = -s \cos \theta$. One recognizes the definition of a Gegenbauer polynomial, hence

$$\varepsilon_\perp^{\mu_1 \cdots \mu_m}(p,q)\varepsilon_{\perp \mu_1 \ldots \mu_m}(p,q') = (-1)^m \frac{m!}{2^m \left(\frac{d-3}{2}\right)_m} \mathcal{C}_m^{(d-3)/2}(\cos \theta). \tag{A.11}$$

Note that our choice of normalization $a_0 = 1$ implies that this is a monic polynomial in $\cos \theta$, i.e. its leading coefficient multiplying $(\cos \theta)^m$ is one. In the case $q' = q$, or $\cos \theta = 1$, we can use

$$\mathcal{C}_m^{(d-3)/2}(1) = \frac{(d-3)_m}{m!}, \tag{A.12}$$

to arrive at

$$\varepsilon_\perp^{\mu_1 \cdots \mu_m}(p,q)\varepsilon_{\perp \mu_1 \ldots \mu_m}(p,q) = (-1)^m \frac{(d-3)_m}{2^m \left(\frac{d-3}{2}\right)_m}. \tag{A.13}$$

## A.2 Non-transverse polarization tensors

By their definition (2.15), the non-transverse polarization tensors satisfy

$$z_{\mu_1} \cdots z_{\mu_\ell} \varepsilon_m^{\mu_1 \ldots \mu_\ell}(p,q) = z_{\mu_1} \cdots z_{\mu_m} \varepsilon_\perp^{\mu_1 \ldots \mu_m} \left(\frac{z \cdot p}{\sqrt{s}}\right)^{\ell-m}. \tag{A.14}$$

The result (2.19) follows straightforwardly from eq. (A.7).

What remains to be proved are the normalization property (2.21) and the identity (2.23). To do so, we construct an explicit representation for the non-transverse tensor using the ansatz

$$\varepsilon_m^{\mu_1 \ldots \mu_\ell}(p,q) = \sum_{j=0}^{\lfloor (\ell-m)/2 \rfloor} b_j \frac{\varepsilon_\perp^{(\mu_1 \ldots \mu_m}(p,q)\eta^{\mu_{m+1}\mu_{m+2}} \cdots \eta^{\mu_{m+2j-1}\mu_{m+2j}} p^{\mu_{m+2j+1}} \cdots p^{\mu_\ell)}}{s^{(\ell-m)/2-j}}, \tag{A.15}$$

where $b_0 = 1$ and the $b_j$ with $j > 0$ are to be determined. The trace of this tensor obeys

$$\eta_{\mu_{\ell-1}\mu_\ell}\varepsilon_m^{\mu_1 \ldots \mu_\ell}(p,q)$$
$$= \sum_{j=0}^{\lfloor (\ell-m)/2 \rfloor - 1} \frac{2(j+1)(d-4+2\ell-2j)b_{j+1} + (\ell-m-2j)(\ell-m-2j-1)b_j}{\ell(\ell-1)}$$
$$\times \frac{\varepsilon_\perp^{(\mu_1 \ldots \mu_m}(p,q)\eta_\perp^{\mu_{m+1}\mu_{m+2}} \cdots \eta_\perp^{\mu_{m+2j-1}\mu_{m+2j}} p^{\mu_{m+2j+1}} \cdots p^{\mu_{\ell-2})}}{s^{(\ell-m)/2-j-1}}. \tag{A.16}$$

Therefore it vanishes if and only if the recursion relation

$$b_{j+1} = -\frac{(\ell-m-2j)(\ell-m-2j-1)}{2(j+1)(d-4+2\ell-2j)} b_j, \tag{A.17}$$

is satisfied. This is solved by

$$b_j = \frac{(\ell-m)!}{j!(\ell-m-2j)!} \frac{(-1)^j}{2^{2j} \left(\frac{d-2}{2} + \ell - j\right)_j}. \tag{A.18}$$

This explicit representation for $\varepsilon_m$ can be used to prove a very useful identity: if we contract one of its indices with $p$, we obtain

$$\frac{p_{\mu_\ell}}{\sqrt{s}} \varepsilon_m^{\mu_1 \ldots \mu_\ell}(p,q) = \sum_{j=0}^{\lfloor (\ell-m-1)/2 \rfloor} \frac{2(j+1)b_{j+1} + (\ell-m-2j)b_j}{\ell}$$
$$\times \frac{\varepsilon_\perp^{(\mu_1 \ldots \mu_m}(p,q)\eta^{\mu_{m+1}\mu_{m+2}} \cdots \eta^{\mu_{m+2j-1}\mu_{m+2j}} p^{\mu_{m+2j+1}} \cdots p^{\mu_{\ell-1})}}{s^{(\ell-m-1)/2-j}}. \tag{A.19}$$

Plugging in the value of $b_j$ obtained above, one arrives at

$$\frac{p_{\mu_\ell}}{\sqrt{s}}\varepsilon_m^{\mu_1\dots\mu_\ell}(p,q) = \frac{(\ell-m)(d-3+\ell+m)}{\ell(d-4+2\ell)}\varepsilon_m^{\mu_1\dots\mu_{\ell-1}}(p,q).\tag{A.20}$$

The fact that the right-hand side is proportional to the polarization tensor with one less unit of spin $\ell$ but with the same $m$ could have been expected, as this is the unique traceless symmetric tensor with transversality properties that can be inferred from the left-hand side. But the proportionality factor is not trivial, and only an explicit computation could determine it. Identity (A.20) is at the core of the derivation of eq. (2.23), and it will also be used below in the computation of the 2-point function.

Using the definition (2.15) together with the symmetry and tracelessness of $\varepsilon_m$, one can write

$$\varepsilon_m^{\mu_1\dots\mu_\ell}(p,q)\varepsilon_{m'\mu_1\dots\mu_\ell}(p,q') = \varepsilon_\perp^{\mu_1\dots\mu_m}(p,q)\frac{p^{\mu_{m+1}}\cdots p^{\mu_\ell}}{s^{(\ell-m)/2}}\varepsilon_{m'\mu_1\dots\mu_\ell}(p,q').\tag{A.21}$$

The right-hand side can then be simplified applying the identity (A.20) recursively, leading to

$$\varepsilon_m^{\mu_1\dots\mu_\ell}(p,q)\varepsilon_{m'\mu_1\dots\mu_\ell}(p,q') = \frac{m!(\ell-m')!}{\ell!(m-m')!}\frac{(d-2+m+m')_{\ell-m}}{2^{\ell-m}\left(\frac{d-2}{2}+m\right)_{\ell-m}}\varepsilon_\perp^{\mu_1\dots\mu_m}(p,q)\varepsilon_{m'\mu_1\dots\mu_m}(p,q').\tag{A.22}$$

Now this expression is zero unless $m' = m$, since any $p^\mu$ or $\eta^{\mu\nu}$ contracted with $\varepsilon_\perp$ vanishes, thus

$$\begin{aligned}
\varepsilon_m^{\mu_1\dots\mu_\ell}&(p,q)\varepsilon_{m'\mu_1\dots\mu_\ell}(p,q')\\
&= \delta_{m'm}\frac{m!(\ell-m)!}{\ell!}\frac{(d-2+2m)_{\ell-m}}{2^{\ell-m}\left(\frac{d-2}{2}+m\right)_{\ell-m}}\varepsilon_\perp^{\mu_1\dots\mu_m}(p,q)\varepsilon_{\perp\mu_1\dots\mu_m}(p,q')\\
&= \delta_{m'm}(-1)^m\frac{m!(\ell-m)!}{\ell!}\frac{(d-2+2m)_{\ell-m}}{2^{\ell-m}\left(\frac{d-2}{2}+m\right)_{\ell-m}}\frac{m!}{2^m\left(\frac{d-3}{2}\right)_m}C_m^{(d-3)/2}(\cos\theta),\quad\text{(A.23)}
\end{aligned}$$

where we have used eq. (A.11) to obtain the last equality. Defining the normalization constant as in eq. (2.22), this completes the proof of the identity (2.23), and subsequently of (2.21) in the case $q' = q$ corresponding to $\cos\theta = 1$.

## A.3  Two-point function

Finally, we give a proof that eq. (2.27) follows from eq. (2.24). As a starting point, we use that fact that the $\varepsilon_m^{\mu_1\dots\mu_\ell}(p,q)$ form a complete basis of traceless symmetric tensors constructed out of the momenta $p$ and $q$. There are $\ell+1$ possible combinations of two vectors $p$ and $q$ into a symmetric tensor with rank $\ell$. The tracelessness condition is achieved by adding or subtracting terms involving the metrics $\eta^{\mu\nu}$, but this does not affect the counting. We gave an explicit construction of $\ell+1$ tensors $\varepsilon_m$ with $m = 0,\dots,\ell$, and showed that they are orthogonal in $m$: therefore they form a basis.

With a complete basis of traceless symmetry tensors, we must be able to write

$$\langle\!\langle\mathcal{O}^{\mu_1\dots\mu_\ell}(-p)\mathcal{O}^{\nu_1\dots\nu_\ell}(p)\rangle\!\rangle = s^{\Delta-d/2}\sum_{m,m'=0}^{\ell}M_{m'm}\varepsilon_{m'}^{\mu_1\dots\mu_\ell}(p,q)\varepsilon_m^{\nu_1\dots\nu_\ell}(p,q),\tag{A.24}$$

where $M_{m'm}$ is a matrix to be determined. This equality is to be understood in the space spanned by $p$ and $q$, i.e. with the replacement (2.26). Using the orthogonality (2.21) of the polarization tensors, we deduce that

$$M_{m'm} = \frac{(-1)^{m'+m}}{\mathcal{N}_{\ell,m}\mathcal{N}_{\ell,m'}}s^{d/2-\Delta}\varepsilon_{m'}^{\mu_1\dots\mu_\ell}(p,q)\varepsilon_m^{\nu_1\dots\nu_\ell}(p,q)\langle\!\langle\mathcal{O}_{\mu_1\dots\mu_\ell}(-p)\mathcal{O}_{\nu_1\dots\nu_\ell}(p)\rangle\!\rangle.\tag{A.25}$$

With the definition (2.24) of the 2-point function, this becomes

$$
M_{m'm} = \frac{(-1)^{m'+m} B_{\Delta,\ell}}{\mathcal{N}_{\ell,m}\mathcal{N}_{\ell,m'}} \sum_{n=0}^{\ell} \frac{(-1)^n 2^n \ell!}{n!(\ell-n)!} \frac{\left(\frac{d}{2}-\Delta\right)_n}{(2-\Delta-\ell)_n} \varepsilon_{m'}^{\mu_1\dots\mu_\ell}(p,q)\varepsilon_m^{\nu_1\dots\nu_\ell}(p,q)
$$
$$
\times \frac{p_{\mu_1}p_{\nu_1}\cdots p_{\mu_n}p_{\nu_n}\eta_{\mu_{n+1}\nu_{n+1}}\cdots\eta_{\mu_\ell\nu_\ell}}{s^n} . \tag{A.26}
$$

Note that we could drop the trace terms and need not average over permutations of indices inside the sum. We can now make use once again of the identity (A.20) to contract all instances of $p$ with the polarization tensors, after which we get

$$
M_{m'm} = \frac{(-1)^{m'+m} B_{\Delta,\ell}}{\mathcal{N}_{\ell,m}\mathcal{N}_{\ell,m'}} \sum_{n=0}^{\ell} \frac{(-1)^n (\ell-n)!(\ell-m)!(\ell-m')!}{2^n \ell! n!(\ell-n-m)!(\ell-n-m')!} \frac{\left(\frac{d}{2}-\Delta\right)_n}{(2-\Delta-\ell)_n}
$$
$$
\times \frac{(d-2+\ell+m-n)_n(d-2+\ell+m'-n)_n}{\left(\frac{d-2}{2}+\ell-n\right)_n^2}
$$
$$
\times \varepsilon_{m'}^{\mu_1\dots\mu_{\ell-n}}(p,q)\eta_{\mu_1\nu_1}\cdots\eta_{\mu_{\ell-n}\nu_{\ell-n}}\varepsilon_m^{\nu_1\dots\nu_{\ell-n}}(p,q). \tag{A.27}
$$

The product of polarization tensors in the last line vanishes unless $m=m'$, which means that $M_{m'm}$ is diagonal. Note that it also vanishes if $m>\ell-n$. We obtain

$$
M_{m'm} = (-1)^m \delta_{m'm} \frac{B_{\Delta,\ell}}{\mathcal{N}_{\ell,m}^2} \sum_{n=0}^{\ell-m} \frac{(-1)^n (\ell-n)![(\ell-m)!]^2}{2^n \ell! n![(\ell-n-m)!]^2} \frac{\left(\frac{d}{2}-\Delta\right)_n}{(2-\Delta-\ell)_n}
$$
$$
\times \frac{(d-2+\ell+m-n)_n^2}{\left(\frac{d-2}{2}+\ell-n\right)_n^2}\mathcal{N}_{\ell-n,m}
$$
$$
= (-1)^m \delta_{m'm} \frac{B_{\Delta,\ell}}{\mathcal{N}_{\ell,m}} \sum_{n=0}^{\ell-m} \frac{(-1)^n (\ell-m)!}{n!(\ell-n-m)!} \frac{\left(\frac{d}{2}-\Delta\right)_n}{(2-\Delta-\ell)_n} \frac{(d-2+\ell+m-n)_n}{\left(\frac{d-2}{2}+\ell-n\right)_n} . \tag{A.28}
$$

Note that the sum is a generalized hypergeometric function evaluated at argument one,

$$
M_{m'm} = (-1)^m \delta_{m'm} \frac{B_{\Delta,\ell}}{\mathcal{N}_{\ell,m}} \, _3F_2\left(\begin{array}{c} -(\ell-m),\ 3-d-\ell-m,\ \frac{d}{2}-\Delta \\ 2-\frac{d}{2}-\ell,\ 2-\Delta-\ell \end{array} ; 1\right), \tag{A.29}
$$

which by Saalschütz's theorem equates

$$
M_{m'm} = (-1)^\ell \delta_{m'm} \frac{B_{\Delta,\ell}}{\mathcal{N}_{\ell,m}} \frac{(\Delta-\ell-d+2)_{\ell-m}}{(\Delta+m-1)_{\ell-m}} . \tag{A.30}
$$

Using this result in eq. (A.24) with the identity $(-1)^\ell \varepsilon_m^{\mu_1\dots\mu_\ell}(p,q) = \varepsilon_{m'}^{\mu_1\dots\mu_\ell}(-p,-q)$, one obtains eq. (2.27).

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
