# Peer review of "Conformal partial waves in momentum space"

_SciPost Physics, doi:SciPost Phys. 10, 081 (2021)_

## Round 2 · Referee Report · Slava Rychkov · 2021-1-30

Report

This is a nice paper with many nontrivial computations. The main result is a representation of a scalar conformal four point function in momentum space as a sum of explicitly given blocks (which can be thought of as conformal blocks in momentum space). I believe this explicit representation is new.

The paper is a bit terse at times. The author states many results without derivation. It’s not clear if the author has a proof of these formulas or if he just checked them in a few cases and conjectures that they should be generally true. E.g. does he have a general proof of 2.27? It would be responsible to provide such information for all nontrivial combinatorial identities (following the tradition of Dolan and Osborn).

I would also prefer to see more details about the exact matching conditions mentioned but not stated on p.15 which fix the form of the Wightman and time-ordered functions.

I should say that I’m confused about the status of time-ordered three-point functions.
Wightman three point functions are conformally invariant tempered distributions, hence their F.T.’s satisfy conformal Ward identities. Time ordered three point functions involve multiply Wightman three point functions by theta-functions and combining the pieces. Of course when we multiply a distribution by a theta-function it’s no longer obvious that the result is a distribution. Has it been proven that this operation can be done in a way that time ordered three point functions are conformally invariant tempered distributions? The author seems to tacitly assume that this is the case. A more detailed discussion and references concerning this subtlety would be welcome. Ref. [85] mentioned in the conclusions, and our future work to appear soon, only treat the case of Wightman functions, so it cannot be used for justification of the time-ordered case.

A similar worry concerns the CFT optical theorem, Eq. (4.2). Is it affected by the unclear status of time-ordered functions as distributions?

A revision along these lines would improve the paper, and I’d be happy to see it.

In preparing this report I benefitted from the discussions with Jiaxin Qiao.

Requested changes

I don't make any hard requirements but I'd be happy if the author tries to improve the paper along the lines stated in the report.

---

## Round 2 · Referee Report · Anonymous · 2021-2-1

Report

The paper computes relatively explicit expressions for scalar conformal blocks in momentum space in Lorentzian signature. While momentum space is often not very natural in a conformal setting, it does appear useful for some applications. For example, the results of this paper provide explicit expressions for scalar conformal blocks in momentum space as analytic functions of the number of space-time dimensions. In position space such expressions are not known, which makes this result especially interesting.

The paper is well organized and the exposition is, for the most part, very clear. I recommend to publish the paper after a minor revision as detailed below.

I only have a couple requests for clarification of some key points. First of all, it would be good to clarify more explicitly for the reader what determines the exact expression in eq (2.18) (e.g. tracelessness, orthogonality to p, ...). The wording "one can take" right above (2.18) suggests some arbitrariness.

Secondly, is it possible to clarify the meaning of (2.32)? Both equations (2.31) and (2.32) are a bit misleading (as clarified below (2.32)), but in the case of (2.31) one can at least directly interpret it as the restriction of the identity operator to some subspace. It does not seem to be possible to interpret (2.32) in this way, at least not naively: morally, (2.32) is supposed to act on polarizations with vector q' but return polarizations with vector q. Ideally, I would suggest expanding a bit more both on the technical meaning and the derivation of (2.32).

Finally, for the reader's benefit, it would be useful to add a short comment about why no action on spin indices of O appears in (3.3). Or, in a sense, why no "action on O" is present. Right now the reader is referred to [65,66], while the argument is rather short: if I understand correctly, it follows because the << ... >> expectation value can be rewritten as <f(p1) f(p2) O(x=0)>, and K acts trivially on O(x=0).

---

## Round 3 · Referee Report · Slava Rychkov (Referee 1) · 2021-3-16

Report

The revision addressed the bulk of my concerns. Some more concerns were answered in the author's reply (although those remarks did not enter the manuscript). I am recommending the paper for publication in SciPost.

---

## Round 3 · Referee Report · Anonymous (Referee 2) · 2021-3-17

Report

The author has addressed all the points raised in my previous report. I especially appreciate the edits to section (2.4), which I now find much easier to understand. I recommend publication in the present form.

---

## Round 3 · Author Response

I would like to thank both referees for their useful comments. I have incorporated all suggested changes in the new version.

As pointed out by one of the referees (Slava Rychkov), the status of the time-ordered functions in CFT is not fully satisfactory. To get a completely unambiguous definition, one should probably begin with considerations about advanced and retarded commutators. This is mentioned at the bottom of page 17 (unchanged from the first version). In fact, this is an issue that I am currently investigating as part of another project, and while it is not possible for me to develop too much on this rich topic in the current manuscript, I can give you a quick summary of what might be the essential points.

In the case where only two operators are time-ordered, as in the vertex functions appearing in section 3, the time-ordered product can be constructed from Wightman functions using the Jost-Lehmann-Dyson representation. This is a representation for the commutator of two operators. The commutator is certainly a tempered distribution as it is the difference between two Wightman functions. From the JLD representation, one can construct retarded and advanced functions, as well as a time-ordered function that corresponds to the difference between the retarded function and one of the Wightman functions. I believe that this construction gives a precise definition of the time-ordered product of 2 operators, and that it establishes rigorously its domain of analyticity.

For time-ordered products involving more than 2 operators, the problem is more difficult. It might still be possible to give a precise definition starting with R-products and using ideas that date back to the work of Steinmann, but this is certainly not easy. From this perspective, the CFT optical theorem (4.2) is not established on rigorous grounds, but its validity has been tested in several examples (including this work).

---

## Round 3 · List of Changes

• Both referees have pointed out that some of the results in section 2 were lacking proofs. I have added a technical appendix with proofs of eqs. (2.18), (2.19), (2.21), (2.23), (2.27), and reference to this appendix in the text before eqs. (2.18), (2.23) and (2.27).

  • I have remodeled section 2.4: instead of writing the completeness relation as (identity) = (sum over spin eigenstates), which as pointed out by one of the referees was only true in some vaguely defined subspace, I have given a new derivation of eq. (2.38) starting with (2.31). The new derivation is more rigorous and hopefully easier to follow for the reader.

  • On page 12, before eq. (3.3), I have added a footnote that explains why no action on spin indices of O appears in that equation.

  • On page 15 and beginning of p. 16, I have extended the discussion of the Wightman 3-point function, in particular about the matching condition between eq. (3.27) and the results of ref. [64]. I did not extend the discussion of the time-ordered function as it was already more detailed than that of the Wightman function.

---

## Editorial Decision

published